# REValueD: Regularised Ensemble Value-Decomposition for Factorisable Markov Decision Processes

**David Ireland,**[*]  **Giovanni Montana**[*][†]
{david.ireland, g.montana}@warwick.ac.uk

## Abstract

Discrete-action reinforcement learning algorithms often falter in tasks with high-dimensional discrete action spaces due to the vast number of possible actions. A recent advancement leverages value-decomposition, a concept from multi-agent reinforcement learning, to tackle this challenge. This study delves deep into the effects of this value-decomposition, revealing that whilst it curtails the over-estimation bias inherent to Q-learning algorithms, it amplifies target variance. To counteract this, we present an ensemble of critics to mitigate target variance. Moreover, we introduce a regularisation loss that helps to mitigate the effects that exploratory actions in one dimension can have on the value of optimal actions in other dimensions. Our novel algorithm, REValueD, tested on discretised versions of the DeepMind Control Suite tasks, showcases superior performance, especially in the challenging humanoid and dog tasks. We further dissect the factors influencing REValueD's performance, evaluating the significance of the regularisation loss and the scalability of REValueD with increasing sub-actions per dimension.

## 1 Introduction

Deep reinforcement learning (DRL) has emerged as a powerful framework that combines the strengths of deep learning and reinforcement learning (RL) to tackle complex decision-making problems in a wide range of domains. By leveraging deep neural networks to approximate value functions and policies, DRL has driven significant breakthroughs in numerous areas, from robotics (Gu et al., 2017; Kalashnikov et al., 2018; Andrychowicz et al., 2020) to game playing (Mnih et al., 2013; Silver et al., 2016; Vinyals et al., 2017) and autonomous systems (Dosovitskiy et al., 2017; Chen et al., 2017b; Kiran et al., 2021). In particular, the use of deep neural networks as function approximators has allowed existing reinforcement learning algorithms to scale to tasks with continuous states and/or action spaces. Nonetheless, problems featuring high-dimensional, discrete action spaces remains relatively unexplored. In these problems, the action space can be thought of as a Cartesian product of discrete sets, *i.e.* $\mathcal{A} = \mathcal{A}_1 \times ... \times \mathcal{A}_N$. In this context, $\mathcal{A}_i$ represents the $i$th sub-action space, containing $n_i$ discrete (sub-)actions. For convenience, we henceforth refer to a Markov Decision Process (Bellman, 1957, MDP) with such a factorisable action space as a factorisable MDP (FMDP).

Traditional DRL algorithms can quickly become ineffective in high dimensional FMDPs, as these algorithms only recognise atomic actions. In this context, an atomic action is defined as any unique combination of sub-actions, each treated as a singular entity; in an FMDP there are $\prod_{i=1}^{N} n_i$ atomic actions. Due to the combinatorial explosion of atomic actions that must be accounted for, standard algorithms such as Q-learning (Watkins and Dayan, 1992; Mnih et al., 2013) fail to learn in these settings as a result of computational impracticalities.

To address these issues, recent approaches have been proposed that emphasise learning about each sub-action space individually (Tavakoli et al., 2018; Seyde et al., 2022). In particular, the DecQN algorithm proposed by Seyde et al. (2022) utilises a strategy known as *value-decomposition* to learn

---

[*]University of Warwick
[†]Alan Turing Institute

utility values for sub-actions. In their methodology, the utility of each selected sub-action is computed independently of others but learnt in such a way that their mean estimates the Q-value for the global action. This approach is inspired by the *centralised training with decentralised execution* paradigm in multi-agent reinforcement learning (MARL) (Kraemer and Banerjee, 2016), where learning the utility of sub-actions is analogous to learning the value of actions from distinct actors. Using this value-decomposition strategy, the $i$th utility function only needs to learn values for the actions in $\mathcal{A}_i$. Consequently, the total number of actions for which a utility value needs to be learned is just $\sum_{i=1}^{N} n_i$. This makes the task significantly more manageable, enabling traditional value-based methods like Deep Q-learning (Mnih et al., 2013; Hessel et al., 2018) to solve FMDPs. Section 3 provides further details on DecQN.

In this paper, we present two primary methodological contributions. First, we build upon DecQN through a theoretical analysis of the value-decomposition when coupled with function approximation. It is well established that Q-learning with function approximation suffers from an over-estimation bias in the target (Thrun and Schwartz, 1993; Hasselt, 2010). Consequently, we explore how the value-decomposition impacts this bias. We demonstrate that, whilst the DecQN decomposition reduces the over-estimation bias in the target Q-values, it inadvertently increases the variance. We further establish that the use of an ensemble of critics can effectively mitigate this increase in variance, resulting in significantly improved performance.

Second, we introduce a regularisation loss that we aim to minimise alongside the DecQN loss. The loss is motivated by the credit assignment issue common in MARL (Weiß, 1995; Wolpert and Tumer, 1999; Zhou et al., 2020; Gronauer and Diepold, 2022), where an exploratory action of one agent can have a negative influence on the value of the optimal action of another agent. Given the similarities with MARL, this credit assignment issue is also an issue when using value-decomposition in FMDPs. By minimising the regularisation loss we help mitigate the impact that exploratory sub-actions can have on the utility values of optimal sub-actions *post-update* by discouraging substantial changes in individual utility estimates. We achieve this by minimising the Huber loss between the selected sub-action utilities and their corresponding values under the target network.

Our work culminates in an approach we call REValueD: **R**egularised **E**nsemble **Value Decomposition**. We benchmark REValueD against DecQN and Branching Dueling Q-Networks (BDQ) (Tavakoli et al., 2018), utilising the discretised variants of DeepMind control suite tasks (Tunyasuvunakool et al., 2020) used by Seyde et al. (2022) for comparison. The experimental outcomes show that REValueD consistently surpasses DecQN and BDQ across a majority of tasks. Of significant note is the marked outperformance of REValueD in the humanoid and dog tasks, where the number of sub-action spaces is exceedingly high ($N = 21$ and 38, respectively). Further, we perform several ablations on the distinct components of REValueD to evaluate their individual contributions. These include analysing how performance evolves with increasing $n_i$ (*i.e.* the size of $|\mathcal{A}_i|$) and examining the impact of the regularisation loss in enhancing the overall performance of REValueD. These extensive experiments underscore the effectiveness and robustness of our approach in handling high-dimensional, discrete action spaces.

## 2 RELATED WORK

**Single-agent approaches to FMDPs**: Several research endeavors have been devoted to exploring the application of reinforcement learning algorithms in environments characterised by large, discrete action spaces (Dulac-Arnold et al., 2015; Van de Wiele et al., 2020). However, these strategies are primarily designed to handle large action spaces consisting of numerous atomic actions, and they do not directly address the challenges associated with FMDPs. More recently, efforts specifically aimed at addressing the challenges posed by FMDPs have been proposed. These works aim to reason about each individual sub-action independently, leveraging either value-based (Sharma et al., 2017; Tavakoli et al., 2018; 2020; Seyde et al., 2022) or policy-gradient methods (Tang and Agrawal, 2020; Seyde et al., 2021). Some researchers have endeavored to decompose the selection of a global action into a sequence prediction problem, where the global action is viewed as a sequence of sub-actions (Metz et al., 2017; Pierrot et al., 2021). However, these methods necessitate defining the sequence ahead of time, which can be challenging without prior information. Tang et al. (2022) analyse a similar value-decomposition, taking the sum of utilities as opposed to the mean. Their analysis looks at fundamental properties of the decomposition of the Q-value, whereas

in this work we analyse the bias/variance of the learning target when using function-approximation in conjunction with value-decomposition. In Appendix B we analyse the sum value-decomposition theoretically and experimentally as a supplement to the analysis of the DecQN decomposition.

**Multi-agent value-decomposition**: A strong connection exists between FMDPs and MARL, especially with respect to single-agent methods utilising value-decomposition (Tavakoli et al., 2018; 2020; Seyde et al., 2022). Owing to the paradigm of *centralised training with decentralised execution* (Kraemer and Banerjee, 2016), value-decomposition has gained significant popularity in MARL. This paradigm permits agents to learn in a centralised fashion whilst operating in a decentralised manner, resulting in a lot of success in MARL (Sunehag et al., 2017; Rashid et al., 2018; 2020; Du et al., 2022). Our proposed method, REValueD, acts as a regulariser for value-decomposition rather than a standalone algorithm.

**Multi-agent value regularisation**: Some researchers have sought to regularise Q-values in MARL through hysteresis (Matignon et al., 2007; Omidshafiei et al., 2017) and leniency (Panait et al., 2006; Palmer et al., 2017). However, these techniques are primarily designed for *independent learners* (Tan, 1993; Claus and Boutilier, 1998), where each agent is treated as an individual entity and the impact of other agents is considered as environmental uncertainty. Conversely, REValueD offers a more flexible approach that can be applied to various value-decomposition methods, extending its applicability beyond independent learners.

**Ensembles**: The use of function approximators in Q-learning-based reinforcement learning algorithms is well known to result in problems due to the maximisation bias introduced by the $\max$ operator used in the target (Thrun and Schwartz, 1993). To mitigate the effects of this maximisation bias, various studies have proposed the use of ensemble methods (Van Hasselt et al., 2016; Lan et al., 2020; Wang et al., 2021). Moreover, ensembles have been suggested as a means to enhance exploration (Osband et al., 2016; Chen et al., 2017a; Lee et al., 2021; Schäfer et al., 2023) or to reduce variance (Anschel et al., 2017; Chen et al., 2021; Liang et al., 2022). In this work we demonstrate that using an ensemble with the DecQN value-decomposition provably reduces the target variance whilst leaving the over-estimation bias unaffected.

## 3 BACKGROUND

### MARKOV DECISION PROCESSES AND FACTORISABLE MARKOV DECISION PROCESSES

We consider a Markov Decision Process (MDP) as a tuple $(\mathcal{S}, \mathcal{A}, \mathcal{T}, r, \gamma, \rho_0)$ where $\mathcal{S}$ and $\mathcal{A}$ are the state and action spaces, respectively, $\mathcal{T} : \mathcal{S} \times \mathcal{A} \to \mathcal{S}$ the transition function, $r : \mathcal{S} \times \mathcal{A} \to \mathbb{R}$ the reward function, $\gamma$ the discount factor and $\rho_0$ the initial state distribution. The objective is to find a policy $\pi : \mathcal{S} \to [0, 1]$, a state-conditioned distribution over actions, that maximises the expected (discounted) returns, $\mathbb{E}_{\tau \sim (\rho_0, \pi)} \left[ \sum_{t=0}^{\infty} \gamma^t r(s_t, a_t) \right]$, where $\tau = (s_0, a_1, ..., a_{T-1}, s_T)$ is a trajectory generated by the initial state distribution $\rho_0$, the transition function $\mathcal{T}$ and the policy $\pi$.

We define a factorisable MDP (FMDP) as an MDP where the action space can be factorised as $\mathcal{A} = \mathcal{A}_1 \times ... \times \mathcal{A}_N$, where $\mathcal{A}_i$ is a set of discrete (sub-)actions. We refer to $\mathbf{a} = (a_1, ..., a_N)$ as the global action, and individual $a_i$'s as sub-actions. If the policy of an FMDP selects sub-actions independently, it is convenient to consider the policy as $\pi(\mathbf{a}|s) = \prod_{i=1}^{N} \pi_i(a_i|s)$, where $\pi_i$ is the policy of the $i$th sub-action space.

### DECOUPLED Q-NETWORKS

Decoupled Q-Networks (DecQN) were introduced by Seyde et al. (2022) to scale up Deep Q-Networks (Mnih et al., 2013) to FMDPs. Rather than learning a Q-value directly, DecQN learns utility values for each sub-action space. If we let $U^i_{\theta_i}(s, a_i)$ be the $i$th utility function, parameterised by $\theta_i$, then for a global action $\mathbf{a} = (a_1, ..., a_N)$ the Q-value is defined as

$$Q_\theta(s, \mathbf{a}) = \frac{1}{N} \sum_{i=1}^{N} U^i_{\theta_i}(s, a_i) \,, \tag{3.1}$$

where $\theta = \{\theta_i\}_{i=1}^N$. This decomposition allows for efficient computation of the $\arg\max$ operator:

$$\arg\max_{\mathbf{a}} Q(s, \mathbf{a}) = \left(\arg\max_{a_1 \in \mathcal{A}_1} U_{\theta_1}^1(s, a_1), ..., \arg\max_{a_N \in \mathcal{A}_N} U_{\theta_N}^N(s, a_N)\right) .$$

To learn the network parameters $\theta$, the following loss is minimised:

$$\mathcal{L}(\theta) = \frac{1}{|B|} \sum_{(s, \mathbf{a}, r, s') \in B} L(y - Q_\theta(s, \mathbf{a})) , \tag{3.2}$$

where $L$ is the Huber loss, $B$ is a batch sampled from the replay buffer, $y = r + \frac{\gamma}{N} \sum_{i=1}^N \max_{a_i' \in \mathcal{A}_i} U_{\bar{\theta}_i}^i(s', a_i')$ is the Q-learning target, and $\bar{\theta} = \{\bar{\theta}_i\}_{i=1}^N$ correspond to the parameters of the target network.

## 4 METHODOLOGY

### DECQN TARGET ESTIMATION BIAS AND VARIANCE

Considering the well-known issue of positive estimation bias associated with Q-learning with function approximation (Thrun and Schwartz, 1993; Hasselt, 2010), our interest lies in understanding how the DecQN decomposition of the global Q-value impacts this over-estimation bias and the variance in the target. Following the assumptions outlined by Thrun and Schwartz (1993), we propose that the approximated Q-function carries some uniformly distributed noise, defined as $Q_\theta(s, \mathbf{a}) = Q^\pi(s, \mathbf{a}) + \epsilon_{s,\mathbf{a}}$. Here, $Q^\pi$ represents the true Q-function corresponding to policy $\pi$, and $\epsilon_{s,\mathbf{a}}$ are independent, identically distributed (i.i.d) Uniform$(-b, b)$ random variables. We then define the target difference as

$$Z_s^{dqn} \triangleq r + \gamma \max_{\mathbf{a}} Q_\theta(s', \mathbf{a}) - \left(r + \gamma \max_{\mathbf{a}} Q^\pi(s', \mathbf{a})\right) ;$$
$$= \gamma \left(\max_{\mathbf{a}} Q_\theta(s', \mathbf{a}) - \max_{\mathbf{a}} Q^\pi(s', \mathbf{a})\right) . \tag{4.1}$$

We refer to $Z_s^{dqn}$ as the target difference when using a DQN, *i.e.* when no value-decomposition is used.

In terms of the DecQN decomposition of the Q-value, it can be assumed that the uniform approximation error stems from the utilities. Hence, we can define $U_{\theta_i}^i(s, a_i) = U_i^{\pi_i}(s, a_i) + \epsilon_{s,a_i}^i$. Analogous to the previous scenario, $U_i^{\pi_i}$ is the true $i$th utility function for policy $\pi_i$ and $\epsilon_{s,a_i}^i$ are i.i.d Uniform$(-b, b)$ random variables. Using the DecQN decomposition from Equation (3.1) in Equation (4.1) we can write the DecQN target difference as

$$Z_s^{dec} = \gamma \left(\frac{1}{N} \sum_{i=1}^N \max_{a_i \in \mathcal{A}_i} U_{\theta_i}^i(s', a_i) - \frac{1}{N} \sum_{i=1}^N \max_{a_i \in \mathcal{A}_i} U_i^{\pi_i}(s', a_i)\right) . \tag{4.2}$$

This leads us to our first result:

**Theorem 1.** *Given the definitions of $Z_s^{dqn}$ and $Z_s^{dec}$ in Equations (4.1) and (4.2), respectively, we have that:*

1. $\mathbb{E}[Z_s^{dec}] \leq \mathbb{E}[Z_s^{dqn}]$ ;

2. *Var*$(Z_s^{dqn}) \leq$ *Var*$(Z_s^{dec})$ .

The detailed proof can be found in Appendix A. The key takeaway from this result is that, whilst the expected value of the target difference is reduced when using the DecQN decomposition – a beneficial factor for reducing the over-estimation bias typically associated with Q-learning – it also results in an increased variance of the target. This trade-off requires further investigation, particularly with respect to its impact on the stability and robustness of learning. The increased variance may lead to more fluctuation in the utility estimates and cause potential instability in learning, which is a notable challenge.

The proposed methodology for the REValueD approach is characterised by the use of an ensemble of critics to address the higher variance observed under the DecQN decomposition. The ensemble approach uses $K$ critics such that each critic $Q^k(s, \mathbf{a})$ is the mean of the utilities $U^i_{\theta_{i,k}}(s, a_i)$, where $\theta_{i,k}$ denotes the parameters of the $i$th utility in the $k$th critic. Each critic in the ensemble is trained with a target

$$y = r + \frac{\gamma}{N} \sum_{i=1}^{N} \max_{a'_i \in \mathcal{A}_i} \bar{U}^i(s', a'_i) \; ;$$

where $\bar{U}^i(s, a_i) = \frac{1}{K} \sum_{k=1}^{K} U^i_{\bar{\theta}_{i,k}}(s, a_i)$ represents the mean utility across all critics, with $\bar{\theta}_{i,k}$ being the parameters of the target utility networks. By applying this ensemble-based approach in REValueD, we aim to effectively reduce the variance introduced by the DecQN decomposition, thus facilitating more stable and efficient learning.

To show that the ensemble-based approach adopted by REValueD brings the variance down, we define the new target difference that incorporates the ensemble as

$$Z_s^{ens} = \gamma \left( \frac{1}{N} \sum_{i=1}^{N} \max_{a_i \in \mathcal{A}_i} \bar{U}^i(s', a_i) - \frac{1}{N} \sum_{i=1}^{N} \max_{a_i \in \mathcal{A}_i} U_i^{\pi_i}(s', a_i) \right) \; . \tag{4.3}$$

It is important to note that the utility function for each critic in the ensemble is now considered to have some uniformly distributed noise, such that $U^i_{\theta_{i,k}}(s, a_i) = U_i^{\pi_i}(s, a_i) + \epsilon^{i,k}_{s,a_i}$, where $\epsilon^{i,k}_{s,a_i}$ are assumed to be i.i.d random variables following a Uniform$(-b, b)$ distribution. Using this target difference, we present our second result:

**Theorem 2.** *Given the definitions of $Z_s^{dec}$ and $Z_s^{ens}$ from Equations* (4.2) *and* (4.3)*, respectively, we have that*

1. $\mathbb{E}[Z_s^{ens}] = \mathbb{E}[Z_s^{dec}]$ ;

2. $Var(Z_s^{ens}) = \frac{1}{K} Var(Z_s^{dec})$ .

The proof is given in Appendix A. Leveraging an ensemble of critics, REValueD provides an effective means to counteract the increased variance inherent in the DecQN decomposition. whilst the ensemble framework leaves the expected value of the target difference unaltered, it reduces its variance. This is an essential property, as it underpins stability throughout the learning process. A further advantageous feature is that the variance reduction is directly proportional to the ensemble size, denoted by $K$. This grants us a direct means of control over the variance, thus allowing for a more controlled learning process. A detailed analysis of how the ensemble size influences the performance of REValueD is presented in Section 5 and in Appendix G we investigate how the ensemble approach affects the training stability.

REGULARISED VALUE-DECOMPOSITION

FMDPs involve multiple sub-actions being executed simultaneously, with the system offering a single scalar reward as feedback. This creates a situation where an optimal sub-action in one dimension might be detrimentally impacted by exploratory sub-actions taken in other dimensions. Under such circumstances, if we minimise the DecQN loss as per Equation (3.2), the utility for the optimal sub-action could be undervalued due to extrinsic influences beyond its control. This insight paves the way for the introduction of the *regularised* component of REValueD. Given that feedback is exclusively available for the *global action*, a credit assignment issue often arises. A global action can see optimal sub-actions from one dimension coupled with exploratory sub-actions from other dimensions. These exploratory sub-actions could potentially yield a low reward or navigate to a low-value subsequent state. Both consequences adversely affect the utility values of optimal sub-actions within the global action by resulting in an underestimated TD target for the optimal sub-action.

To counteract this impact, we propose the introduction of a regularisation term in the model's update equation, thereby minimising the following loss:

$$\mathcal{L}_a(\theta) = \frac{1}{|B|} \sum_{(s,\mathbf{a},r,s') \in B} \sum_{i=1}^{N} w_i L \left( U_{\bar{\theta}_i}(s, a_i) - U_{\theta_i}(s, a_i) \right) \; ; \tag{4.4}$$

where $w_i = 1 - \exp(-|\delta_i|)$ and $\delta_i = y - U_{\theta_i}(s, a_i)$, with $y$ being defined as in Equation (3.2).

The functional form for the weights is chosen such that as $|\delta_i|$ grows larger, the weights tend to one. The rationale being that, for large $|\delta_i|$, the reward/next state is likely influenced by the effect of other sub-actions. As a result, we want to regularise the update to prevent individual utilities from being too under or overvalued. This is achieved by keeping them in close proximity to the existing values – a process managed by weights that increase as $|\delta_i|$ increases. We offer more insight into the functional form of the weights in Appendix I.

By introducing this regularisation loss, we gain finer control over the impact of an update *at the individual utility level*. Unlike the DecQN loss, which updates the mean of the utility estimates, our approach facilitates direct regulation of the impact of an update on specific utilities. Consequently, the total loss that REValueD aims to minimise is then defined by

$$\mathcal{L}_{tot}(\theta) = \mathcal{L}(\theta) + \beta \mathcal{L}_a(\theta) ; \tag{4.5}$$

where $\beta$ acts as hyper-parameter determining the extent to which we aim to minimise the regularisation loss. Throughout our experiments, we keep $\beta$ at a consistent value of $0.5$, and we perform an ablation on the sensitivity to $\beta$ in Appendix D. It is worth noting that although we introduce the regularisation loss for a single critic (*i.e.* $K = 1$) for ease of notation, its extension to an ensemble of critics is straightforward.

## 5 EXPERIMENTS

In this Section we benchmark REValueD on the discretised versions of the DeepMind Control Suite tasks (Tunyasuvunakool et al., 2020), as utilised by Seyde et al. (2022). These tasks represent challenging control problems, and when discretised, they can incorporate up to 38 distinct sub-action spaces. We also provide results for a selection of discretised MetaWorld tasks (Yu et al., 2020) in Appendix J. Finally, as an additional baseline, we compare REValueD to a version of DecQN with a distributional critic in Appendix L.

Unless stated otherwise, we employ the same Bang-Off-Bang discretisation as Seyde et al. (2022), in which each dimension of the continuous action is discretised into three bins. For comparative analysis, we evaluate REValueD against the vanilla DecQN (Seyde et al., 2022) and a Branching Dueling Q-Network (BDQ) (Tavakoli et al., 2018). For a more like-to-like comparison, we compare REValueD to an ensembled variant of BDQ in Appendix K in select DMC tasks. To measure the performance, after every 1000 updates we plot the returns of a test episode where actions are selected according to a greedy policy. Implementation details are given in Appendix C.

DEEPMIND CONTROL SUITE

The mean performance, together with an accompanying $95\%$ confidence interval, is shown in Figure 1. This comparison includes the results achieved by REValueD, DecQN, and BDQ on the DM Control Suite tasks. It is immediately clear that BDQ is the least successful, with a complete failure to learn in the humanoid environments. We also observe that REValueD demonstrates considerable improvement over DecQN in most tasks, especially the more challenging tasks where the number of sub-action spaces is larger. This advantage is particularly noticeable in the humanoid and dog tasks, which have an exceptionally large number of sub-action spaces. These findings provide strong evidence for the increased learning efficiency conferred by REValueD in FMDPs.

The performance of these algorithms as the number of sub-actions per dimension increases is another important factor to consider. We investigate this by varying the number of bins used to discretise the continuous actions. In Figure 2 we see the results in the demanding dog-walk task. For $n = 30$, REValueD maintains a strong performance, whereas the performance of DecQN has already started to falter. We can see that even for $n = 75$ or larger, REValueD still demonstrates an acceptable level of performance, whereas DecQN is only just showing signs of learning. Interestingly, BDQ performs consistently in the dog-walk task, despite the large $n_i$ values, managing to learn where DecQN falls short. We present further results in Figure 5 in Appendix E.

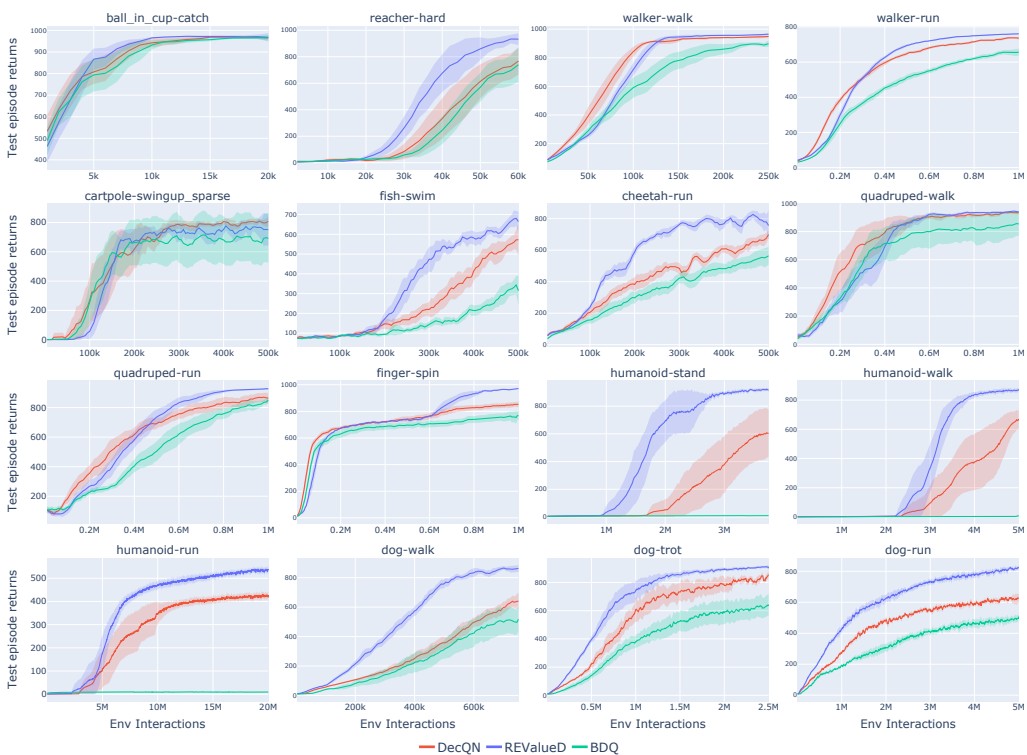

Figure 1: Performance for the Discretised DeepMind Control Suite tasks. We compare REValueD with DecQN and BDQ. The solid line corresponds to the mean of 10 seeds, with the shaded area corresponding to a 95% confidence interval.

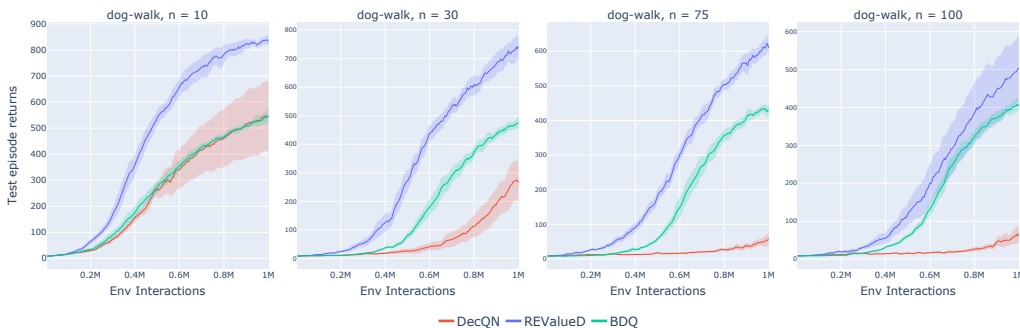

Figure 2: Here we assess how the performances of DecQN, BDQ and REValueD are affected by increasing the size of each sub-action space. We conduct experiments on the fish-swim, cheetah-run and dog-walk tasks. $n$ corresponds to the size of the sub-action spaces, *i.e.* $|\mathcal{A}_i| = n$ for all $i$. The solid line corresponds to the mean of 10 seeds, with the shaded area corresponding to a 95% confidence interval. Further results are given in Figure 5 in Appendix E.

ABLATION STUDIES

**Relative contribution of the regularisation loss:** To demonstrate the effect that the regularisation loss (Equation (4.4)) has on the performance of REValueD we analyse its contribution in the most challenging DM Control Suite tasks. We also provide a further ablation in Appendix H that demonstrates how the loss can delay the negative effects of exploratory updates on optimal sub-action

Table 1: This Table demonstrates the impact of the regularisation loss by contrasting the performance of REValueD with DecQN equipped only with an ensemble. Our comparison leverages the humanoid and dog tasks, deemed to be the most demanding tasks, thus necessitating careful credit assignment for sub-actions. We report the mean $\pm$ standard error of 10 seeds.

| Task | Algorithm | | |
|---|---|---|---|
| | DecQN | DecQN+Ensemble | REValueD |
| Humanoid-Stand | $604.82 \pm 85.1$ | $832.99 \pm 11.3$ | $\mathbf{915.80 \pm 6.12}$ |
| Humanoid-Walk | $670.62 \pm 34.1$ | $817.63 \pm 7.66$ | $\mathbf{874.33 \pm 3.63}$ |
| Humanoid-Run | $416.81 \pm 8.77$ | $478.11 \pm 4.59$ | $\mathbf{534.81 \pm 10.8}$ |
| Dog-Walk | $641.13 \pm 28.8$ | $819.95 \pm 22.0$ | $\mathbf{862.31 \pm 12.0}$ |
| Dog-Trot | $856.48 \pm 12.2$ | $878.47 \pm 6.33$ | $\mathbf{902.01 \pm 7.65}$ |
| Dog-Run | $625.68 \pm 12.5$ | $750.73 \pm 11.2$ | $\mathbf{821.17 \pm 8.10}$ |

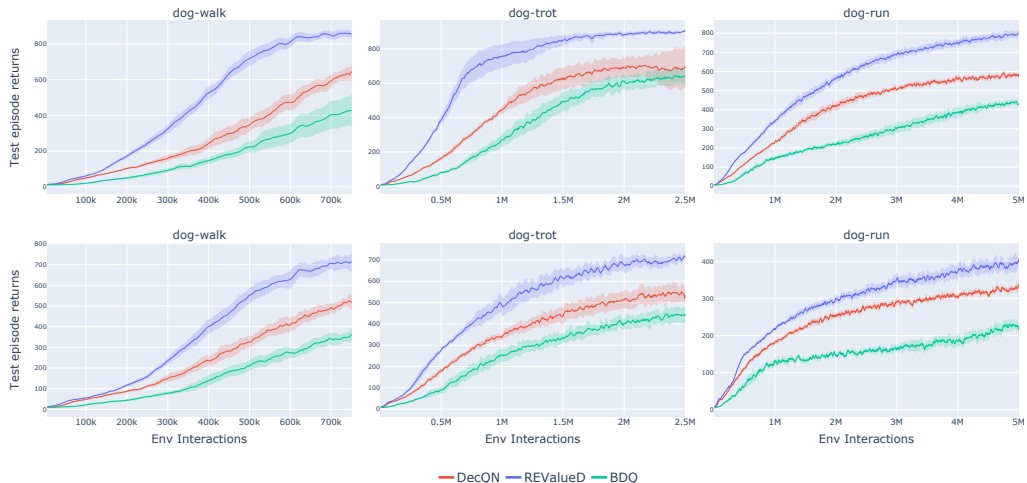

Figure 3: Stochastic environment tasks. In the top row we added Gaussian white noise ($\sigma = 0.1$) to the rewards, whilst in the bottom row we added Gaussian white noise to the state. Further results are given in Figure 6 in Appendix F.

utility values in a Tabular FMDP. Table 1 compares the performance of REValueD both with and without the regularisation loss; the latter we refer to as DecQN+Ensemble. As demonstrated in the Table, the inclusion of an ensemble of critics noticeably enhances the performance of DecQN. Further, the incorporation of the regularisation loss further augments the performance of the ensemble, clearly illustrating the merits of the regularisation loss.

**Stochastic Environments:** Here we extend our analysis to stochastic variants of a selection of the DM Control Suite tasks. Stochastic environments exacerbate the credit assignment problem outlined in Section 4 as there is now an extra source of uncertainty that each decoupled actor must contend with. We consider two types of stochasticity, adding Gaussian white noise to the reward and states, respectively. Figure 3 shows the results in the three variants of the dog task. We observe that all three algorithms are robust to stochasticity being added to the rewards with the performance being largely the same in the three dog tasks. When stochasticity is added to the states, the performance of all three algorithms drops, though we see that the same hierarchy of performance is maintained, *i.e.* REValueD outperforms DecQN whilst both outperform BDQ. In Appendix F we show show similar results in more stochastic DM Control Suite environments.

**Assessing the impact of the ensemble size:** Of particular interest is how the size of the ensemble used in REValueD affects performance. In Table 2 we provide the final asymptotic results. In gen-

Table 2: Asymptotic results for REValueD with varying ensemble size across various DM Control Suite tasks. We report the mean $\pm$ standard error over 10 seeds.

| Task | Ensemble Size | | | | | |
|---|---|---|---|---|---|---|
| | 1 | 3 | 5 | 10 | 15 | 20 |
| Walker-Run | $763.97 \pm 4.75$ | $\mathbf{779.40 \pm 1.29}$ | $769.25 \pm 2.59$ | $761.74 \pm 2.30$ | $762.32 \pm 4.95$ | $755.86 \pm 7.35$ |
| Cheetah-Run | $759.01 \pm 12.5$ | $843.46 \pm 16.3$ | $781.30 \pm 14.3$ | $828.60 \pm 25.9$ | $\mathbf{883.55 \pm 2.99}$ | $807.75 \pm 9.94$ |
| Quadruped-Run | $911.01 \pm 13.4$ | $\mathbf{927.34 \pm 5.14}$ | $927.32 \pm 8.17$ | $924.52 \pm 3.92$ | $926.79 \pm 4.27$ | $922.33 \pm 7.56$ |
| Dog-Walk | $838.39 \pm 8.40$ | $\mathbf{914.51 \pm 4.01}$ | $904.47 \pm 2.11$ | $886.73 \pm 7.38$ | $899.77 \pm 7.90$ | $878.40 \pm 8.81$ |
| Dog-Trot | $843.61 \pm 19.8$ | $915.67 \pm 4.14$ | $912.97 \pm 5.09$ | $910.24 \pm 10.4$ | $\mathbf{915.63 \pm 3.59}$ | $897.34 \pm 5.16$ |
| Dog-Run | $675.51 \pm 9.10$ | $771.25 \pm 11.6$ | $815.29 \pm 15.9$ | $821.17 \pm 8.10$ | $\mathbf{834.77 \pm 4.16}$ | $823.20 \pm 7.29$ |

eral, the performance is reasonably robust to the ensemble size for the walker/cheetah/quadruped-run tasks. In particular, we note that an ensemble size of 3 in walker-run has the best asymptotic performance. It has been noted that some target variance can be useful for exploration purposes (Chen et al., 2021), and so it is possible that for an easier task like walker using a higher ensemble reduces the variance too much. However, as the complexity of the task increases we start to see the beneficial impact that a larger ensemble size has. This is evident in the demanding dog-run tasks, where performance peaks with an ensemble size of 15. To summarise these findings, it may be beneficial to use a smaller ensemble size for tasks which are easier in nature. A larger ensemble size improves performance for more complex tasks.

## 6    CONCLUSION

Recent works have successfully taken the idea of value-decomposition from MARL and applied it to single agent FMDPs. Motivated by the known issues surrounding maximisation bias in Q-learning, we analyse how this is affected when using the DecQN value-decomposition. We show theoretically that whilst the estimation bias is lowered, the target variance increases. Having a high target variance can cause instabilities in learning. To counteract this, we introduce an ensemble of critics to control the target variance, whilst showing that this leaves the estimation bias unaffected.

Drawing on the parallels between single agent FMDPs and MARL, we also address the credit assignment issue. Exploratory sub-actions from one dimension can effect the value of optimal sub-actions from another dimension, giving rise to a credit assignment problem. If the global action contained some exploratory sub-actions then the TD-target can be undervalued with respect to optimal sub-actions in the global action. To counteract this, we introduce a regularisation loss that we aim to minimise alongside the DecQN loss. The regularisation works at the individual utility level, mitigating the effect of exploratory sub-actions by enforcing that utility estimates do not stray too far from their current values.

These two contributions lead to our proposed algorithm: REValueD. We compare the performance of REValueD to DecQN and BDQ in tasks from the DM Control Suite. We find that, in most tasks, REValueD greatly outperforms DecQN and BDQ, notably in the challenging humanoid and dog tasks which have $N = 21$ and $38$ sub-action spaces, respectively. We also provide various ablation studies, including the relative contribution that the regularisation loss has on the overall performance of REValueD and how the performance changes as the number of sub-actions per dimension increases.

Potential avenues for future work include a more rigorous exploration of the work in Appendix L to better understand the benefits of distributional reinforcement learning (Bellemare et al., 2023) in FMDPs, as a distributional perspective to learning can help deal with the uncertainty induced by exploratory sub-actions. Further, using an uninformative $\epsilon$-greedy exploration strategy can be wasteful in FMDPs due to the many possible combinations of sub-actions, and so more advanced exploration techniques could be developed to effectively choose optimal sub-actions in one dimension whilst continuing to explore in other dimensions.

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

## A   PROOF OF THEOREMS 1 AND 2

**Lemma 1.** *Let $\{X_i\}_{i=1}^n$ be i.i.d Uniform$(-b, b)$ random variables, and let $Z = \max_i\{X_i\}_{i=1}^n$ be their maximum. Then, $\mathbb{E}[Z] = b\frac{n-1}{n+1}$ and Var$(Z) = \frac{4b^2 n}{(n+1)^2(n+2)}$.*

*Proof.* Let $F_X(x) = \frac{x+b}{2b}$ be the CDF of a random variable with distribution Uniform$(-b, b)$. Thus, we have that

$$
\begin{aligned}
F_Z(z) &= \mathbb{P}(Z \leq z) \\
&= \mathbb{P}(\max_i\{X_i\}_{i=1}^n \leq z) \\
&= [F_X(z)]^n \\
&= \left[\frac{z+b}{2b}\right]^n .
\end{aligned}
$$

Now, the pdf of $Z$ is given by $f_Z(z) = \frac{\mathrm{d}}{\mathrm{d}z}F_Z(z) = \frac{n}{2b}\left(\frac{z+b}{2b}\right)^{n-1}$. From this, we can deduce that

$$
\mathbb{E}\left[Z\right] = \int_{-b}^{b} z\frac{n}{2b}\left(\frac{z+b}{2b}\right)^{n-1} \mathrm{d}z = b\frac{n-1}{n+1} ;
$$

$$
\mathrm{Var}(Z) = \left(\int_{-b}^{b} z^2\frac{n}{2b}\left(\frac{z+b}{2b}\right)^{n-1} \mathrm{d}z\right) - \mathbb{E}\left[Z\right]^2 = \frac{4b^2 n}{(n+1)^2(n+2)} .
$$

Note that the first integral can be solved by using the substitution $y = \frac{z+b}{2b}$ and the second can be solved by parts followed by a similar substitution. □

**Lemma 2.** *For $n_i \in \mathbb{N}, n_i \geq 2$, we have $\frac{1}{N}\sum_{i=1}^{N}\frac{n_i-1}{n_i+1} \leq \frac{\left(\prod_{i=1}^{N} n_i\right)-1}{\left(\prod_{i=1}^{N} n_i\right)+1}$ for $N \geq 1$.*

*Proof.* Note that $f(x) = \frac{x-1}{x+1}$ is concave and increasing on $\mathbb{R}_+$. Thus, using Jensen's inequality we have

$$
\frac{1}{N}\sum_{i=1}^{N} f(n_i) \leq f\left(\frac{1}{N}\sum_{i=1}^{N} n_i\right) .
$$

Since $f$ is increasing we also have

$$
f\left(\frac{1}{N}\sum_{i=1}^{N} n_i\right) \leq f\left(\prod_{i=1}^{N} n_i\right) .
$$

We have thus shown that

$$
\frac{1}{N}\sum_{i=1}^{N}\frac{n_i-1}{n_i+1} \leq \frac{\left(\prod_{i=1}^{N} n_i\right)-1}{\left(\prod_{i=1}^{N} n_i\right)+1} .
$$

□

**Lemma 3.** *For $n_i \in \mathbb{N}, n_i \geq 2$, we have $\frac{\prod_{i=1}^{N} n_i}{\left(1+\prod_{i=1}^{N} n_i\right)^2\left(2+\prod_{i=1}^{N} n_i\right)} \leq \frac{1}{N^2}\sum_{i=1}^{N}\frac{n_i}{(n_i+1)^2(n_i+2)}$ for $N \geq 1$.*

*Proof.* Note that $f(x) = \frac{x}{(x+1)^2(x+2)}$ is convex and decreasing on $\{x : x \in \mathbb{R}_+, x \geq 2\}$. Thus, using Jensen's inequality we have

$$
\frac{1}{N} f\left(\frac{1}{N}\sum_{i=1}^{N} n_i\right) \leq \frac{1}{N^2}\sum_{i=1}^{N} f(n_i) .
$$

Now we need to show that $f\left(\prod_{i=1}^N n_i\right) \leq \frac{1}{N} f\left(\frac{1}{N}\sum_{i=1}^N n_i\right)$; that is, we want to show that

$$\frac{\prod_{i=1}^N n_i}{\left(1+\prod_{i=1}^N n_i\right)^2\left(2+\prod_{i=1}^N n_i\right)} \leq \frac{\frac{1}{N}\sum_{i=1}^N n_i}{N\left(1+\frac{1}{N}\sum_{i=1}^N n_i\right)^2\left(2+\frac{1}{N}\sum_{i=1}^N n_i\right)} \ ,$$

or, equivalently, that

$$N\left(1+\frac{1}{N}\sum_{i=1}^N n_i\right)^2\left(1+\frac{2}{\frac{1}{N}\sum_{i=1}^N n_i}\right) \leq \left(1+\prod_{i=1}^N n_i\right)^2\left(1+\frac{2}{\prod_{i=1}^N n_i}\right) \ .$$

Let $g(x) = (1+x)^2\left(1+\frac{2}{x}\right)$. It is straight-forward to verify that $g$ is increasing for $x \geq 2$ and satisfies $g(2x) \geq 2g(x)$ for $x \geq 2$. We have that $\prod_{i=1}^N n_i \geq 2^{N-1}\frac{1}{N}\sum_{i=1}^N n_i$ since all factors are greater than or equal to 2, and at least one factor is greater than or equal to $\frac{1}{N}\sum_{i=1}^N n_i$. It then follows that

$$g\left(\prod_{i=1}^N n_i\right) \geq g\left(2^{N-1}\frac{1}{N}\sum_{i=1}^N n_i\right) \geq 2^{N-1}g\left(\frac{1}{N}\sum_{i=1}^N n_i\right) \geq Ng\left(\frac{1}{N}\sum_{i=1}^N n_i\right) \ ;$$

where in the last step we have used Bernoulli's inequality:

$$2^{N-1} = (1+1)^{N-1} \geq 1 + (N-1) = N \ .$$

We have thus shown that $g\left(\prod_{i=1}^N n_i\right) \geq Ng\left(\frac{1}{N}\sum_{i=1}^N n_i\right)$, and, subsequently, that $f\left(\prod_{i=1}^N n_i\right) \leq \frac{1}{N^2}\sum_{i=1}^N f(n_i)$.

$\square$

**Theorem 1.** *Given the definitions of $Z_s^{dqn}$ and $Z_s^{dec}$ in Equations* (4.1) *and* (4.2)*, respectively, we have that:*

1. $\mathbb{E}[Z_s^{dec}] \leq \mathbb{E}[Z_s^{dqn}]$ ;

2. *Var*$(Z_s^{dqn}) \leq$ *Var*$(Z_s^{dec})$ .

**Proof of Theorem 1**: First note that in Lemmas 2 and 3 we have assumed that $n_i \in \mathbb{N}, n_i \geq 2$. This is because $n_i$ corresponds to the size of a discrete (sub)action space, so we know it must be at least size 2 (otherwise there would be no decision to make).

1. As shown in Thrun and Schwartz (1993), $\mathbb{E}[Z_s^{dqn}] = \gamma b\frac{|\mathcal{A}|-1}{|\mathcal{A}|+1}$. For an FMDP, $|\mathcal{A}| = \left(\prod_{i=1}^N n_i\right)$, from which it follows that $\mathbb{E}[Z_s^{dqn}] = \gamma b\frac{\left(\prod_{i=1}^N n_i\right)-1}{\left(\prod_{i=1}^N n_i\right)+1}$. For the DecQN decomposition, the expectation is

$$\mathbb{E}[Z_s^{dec}] = \frac{\gamma}{N}\sum_{i=1}^N \mathbb{E}\left[\max_{a_i\in\mathcal{A}_i}\epsilon_{s,a_i}^i\right] \ ;$$
$$= \frac{\gamma b}{N}\sum_{i=1}^N \frac{n_i-1}{n_i+1} \ .$$

Here, we have used the expectation from Lemma 1, as the errors are i.i.d Uniform$(-b,b)$ random variables. We have shown in Lemma 2 that $\frac{1}{N}\sum_{i=1}^N \frac{n_i-1}{n_i+1} \leq \frac{\left(\prod_{i=1}^N n_i\right)-1}{\left(\prod_{i=1}^N n_i\right)+1}$, therefore $\mathbb{E}[Z_s^{dec}] \leq \mathbb{E}[Z_s^{dqn}]$.

2. First, since the errors are i.i.d Uniform$(-b, b)$ random variables we use the variance from Lemma 1 to get

$$\text{Var}(Z_s^{dqn}) = \gamma^2 \text{Var}\left(\max_{\mathbf{a}} \epsilon_{s,\mathbf{a}}\right) = \gamma^2 \frac{4b^2 \prod_{i=1}^{N} n_i}{\left(1 + \prod_{i=1}^{N} n_i\right)^2 \left(2 + \prod_{i=1}^{N} n_i\right)} \ ;$$

$$\text{Var}(Z_s^{dec}) = \frac{\gamma^2}{N^2} \sum_{i=1}^{N} \text{Var}\left(\max_{a_i \in \mathcal{A}_i} \epsilon_{s,a_i}^i\right) = \frac{\gamma^2}{N^2} \sum_{i=1}^{N} \frac{4b^2 n_i}{(n_i + 1)^2(n_i + 2)} \ .$$

We have shown in Lemma 3 that $\frac{\prod_{i=1}^{N} n_i}{\left(1 + \prod_{i=1}^{N} n_i\right)^2 \left(2 + \prod_{i=1}^{N} n_i\right)} \leq \frac{1}{N^2} \sum_{i=1}^{N} \frac{n_i}{(n_i + 1)^2(n_i + 2)}$, therefore $\text{Var}(Z_s^{dqn}) \leq \text{Var}(Z_s^{dec})$.

**Theorem 2.** *Given the definitions of $Z_s^{dec}$ and $Z_s^{ens}$ from Equations (4.2) and (4.3), respectively, we have that*

1. $\mathbb{E}[Z_s^{ens}] = \mathbb{E}[Z_s^{dec}]$ ;

2. $Var(Z_s^{ens}) = \frac{1}{K} Var(Z_s^{dec})$ .

**Proof of Theorem 2**

1. Since for fixed $i$ the errors are i.i.d. across $k$, and in fact have the same distribution as the error terms in the DecQN decomposition, we have that

$$\mathbb{E}[Z_s^{ens}] = \frac{\gamma}{N} \sum_{i=1}^{N} \frac{1}{K} \sum_{k=1}^{K} \mathbb{E}\left[\max_{a_i \in \mathcal{A}_i} \epsilon_{s,a_i}^{i,k}\right] \ ;$$

$$= \frac{\gamma}{N} \sum_{i=1}^{N} \frac{1}{K} \cdot K \mathbb{E}\left[\max_{a_i \in \mathcal{A}_i} \epsilon_{s,a_i}^i\right] \ ;$$

$$= \frac{\gamma}{N} \sum_{i=1}^{N} \mathbb{E}\left[\max_{a_i \in \mathcal{A}_i} \epsilon_{s,a_i}^i\right] \ ;$$

$$= \mathbb{E}[Z_s^{dec}] \ .$$

2. Using a similar argument to 1., we have that

$$\text{Var}(Z_s^{ens}) = \frac{\gamma^2}{N^2} \sum_{i=1}^{N} \frac{1}{K^2} \sum_{k=1}^{K} \text{Var}\left(\max_{a_i \in \mathcal{A}_i} \epsilon_{s,a_i}^{i,k}\right) \ ;$$

$$= \frac{\gamma^2}{N^2} \sum_{i=1}^{N} \frac{1}{K^2} \cdot K \text{Var}\left(\max_{a_i \in \mathcal{A}_i} \epsilon_{s,a_i}^i\right) \ ;$$

$$= \frac{1}{K} \left[\frac{\gamma^2}{N^2} \sum_{i=1}^{N} \text{Var}\left(\max_{a_i \in \mathcal{A}_i} \epsilon_{s,a_i}^i\right)\right] \ ;$$

$$= \frac{1}{K} \text{Var}(Z_s^{dec}) \ .$$

## B  ANALYSIS OF THE SUM VALUE-DECOMPOSITION

Tang et al. (2022) propose a value-decomposition similar to the decomposition proposed by Seyde et al. (2022) (Equation (3.1)). Their value-decomposition approximates the global Q-value as the *sum* of the utilities, whereas DecQN define their decomposition as the *mean* of the utilities. Using

our notation, the value-decomposition proposed by Tang et al. (2022) is defined as

$$Q_\theta(s, \mathbf{a}) = \sum_{i=1}^{N} U_{\theta_i}^i(s, a_i) \ .$$

Using this decomposition in Equation (4.1) we can write the sum target difference as

$$Z_s^{sum} = \gamma \left( \sum_{i=1}^{N} \max_{a_i \in \mathcal{A}_i} U_{\theta_i}^i(s', a_i) - \sum_{i=1}^{N} \max_{a_i \in \mathcal{A}_i} U_i^{\pi_i}(s', a_i) \right) \ . \tag{B.1}$$

Whilst the decompositions are similar, we will now show that under this value-decomposition the expected value and the variance of this target difference are higher than that of the DQN target difference.

**Theorem 3.** *Given the definitions of $Z_s^{dqn}$ and $Z_s^{sum}$ in Equations (4.1) and (B.1), respectively, we have that:*

1. $\mathbb{E}[Z_s^{dqn}] \leq \mathbb{E}[Z_s^{sum}]$ ;

2. $Var(Z_s^{dqn}) \leq Var(Z_s^{sum})$ .

**Proof of Theorem 3**

1. We can see that $\mathbb{E}[Z_s^{sum}] = N\mathbb{E}[Z_s^{dec}] = \gamma b \sum_{i=1}^{N} \frac{n_i-1}{n_i+1}$, so it remains to be shown that $\sum_{i=1}^{N} \frac{n_i-1}{n_i+1} \geq \frac{(\prod_{i=1}^{N} n_i)-1}{(\prod_{i=1}^{N} n_i)+1}$. First, let $f(x) = \frac{x-1}{x+1}$ and note that $\lim_{x\to\infty} f(x) = 1$ and that $f$ is increasing for $x \geq 2$. Since $f$ is increasing, and recalling that $n_i \geq 2$, for $N \geq 1$ we have

$$\sum_{i=1}^{N} f(n_i) \geq \sum_{i=1}^{N} f(2) \ ;$$
$$= \sum_{i=1}^{N} \frac{1}{3} = \frac{N}{3} \ .$$

Now, $f\left(\prod_{i=1}^{N} n_i\right) \leq 1 \leq \frac{N}{3}$ for $N \geq 3$ and the case for $N = 1$ is trivial. For $N = 2$ we want to show that

$$\frac{n_1 n_2 - 1}{n_1 n_2 + 1} \leq \frac{n_1 - 1}{n_1 + 1} + \frac{n_2 - 1}{n_2 + 1} \ .$$

Re-arranging this term, we get

$$\frac{(n_1 n_2 - 1)(n_1 - 1)(n_2 - 1)}{(n_1 + 1)(n_2 + 1)(n_1 n_2 + 1)} \geq 0 \ ;$$

which is clearly true for $n_1, n_2 \geq 2$, and so the inequality also holds for $N = 2$.

2. We can see that $Var(Z_s^{sum}) = N^2 var(Z_s^{dec}) = \gamma^2 \sum_{i=1}^{N} \frac{4b^2 n_i}{(n_i+1)^2 (n_i+2)}$. As we have shown that $Var(Z_s^{dec}) \geq Var(Z_s^{dqn})$ it follows immediately that $Var(Z_s^{sum}) \geq Var(Z_s^{dqn})$.

Theorem 3 tells us that, when using function approximators, decomposing the Q-function using the sum of utilities leads to a higher bias in the target value compared to no decomposition. This is in contrast to the DecQN value-decomposition, which reduces the bias. However, both decompositions come at the cost of increased variance – though we show as part of the proof that the sum decomposition also has a higher variance than the DecQN decomposition.

Experimentally, we compare REValueD and DecQN with the sum value-decomposition (which we name DecQN-Sum) in Figure 4. We can see from the Figure that using the sum value-decomposition generally results in poorer performance than the mean. The poor performance of DecQN-Sum compared to DecQN and REValueD can likely be attributed to our findings in Theorem 3, that the sum value-decomposition increases the overestimation bias compared to using the mean as in

DecQN. In Appendix K we offer a further comparison of REValueD with DecQN-Sum equipped with an ensemble for a fairer comparison.

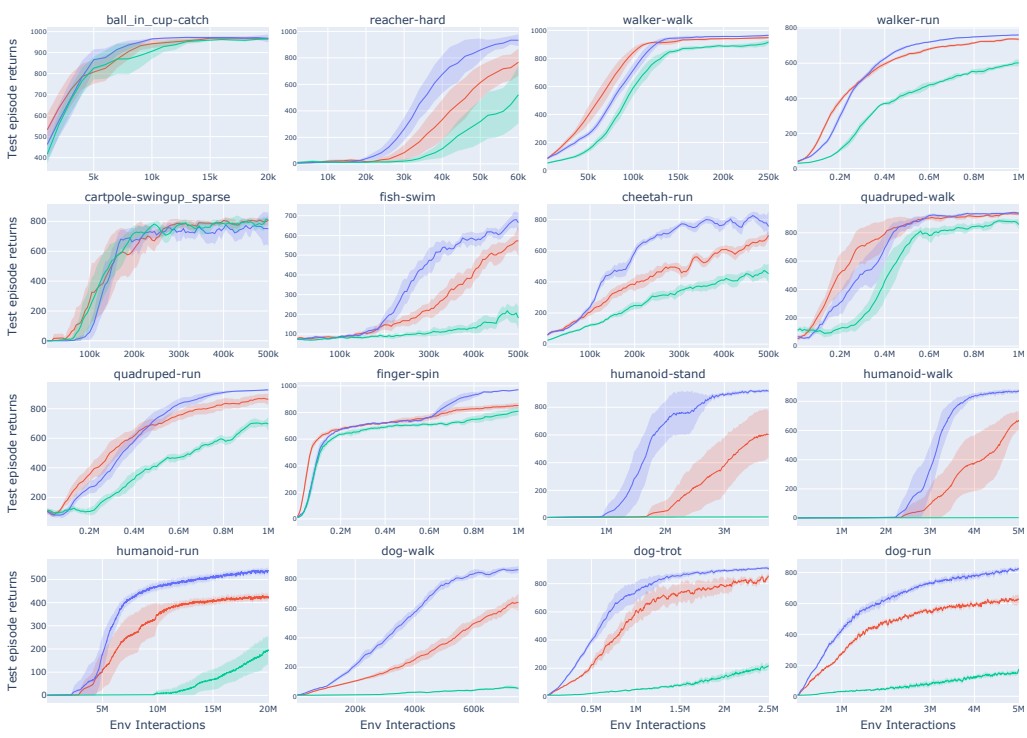

Figure 4: Further results comparing REValueD and DecQN to DecQN using the sum value-decomposition (DecQN-Sum). The solid line corresponds to the mean of 10 seeds, with the shaded area corresponding to a 95% confidence interval.

## C EXPERIMENT DETAILS

**Hyperparameters:** We largely employ the same hyperparameters as the original DecQN study, as detailed in Table 4, along with those specific to REValueD. Exceptions include the decay of the exploration parameter ($\epsilon$) to a minimum value instead of keeping it constant, and the use of Polyak-averaging for updating the target network parameters, as opposed to a hard reset after every specified number of updates. We maintain the same hyperparameters across all our experiments. During action selection in REValueD, we follow a deep exploration technique similar to that proposed by Osband et al. (2016), where we sample a single critic from the ensemble at each time-step during training and follow an $\epsilon$-greedy policy based on that critic's utility estimates. For test-time action selection, we average over the ensemble and then act greedily according to the mean utility values.

**Architecture and implementation details:** As in the original DecQN study, our architecture consists of a fully connected network featuring a residual block followed by layer normalisation. The output from this sequence is fed through a fully connected linear layer predicting value estimates for each sub-action space. In implementing the ensemble for REValueD, we follow the approach given by Tarasov et al. (2022), which has been demonstrated to have minor slowdowns for relatively small ensemble sizes (Beeson and Montana, 2024, Table 8).

In Table 3 we report the mean time taken for DecQN and REValueD, in minutes, when run on the same machine. Each task was run for 500k environment interactions (100k updates). We can see from the table REValueD takes around 25% longer than DecQN, though it is important to note that REValueD is more sample efficient than DecQN so less environment interactions are needed to achieve a superior performance.

Table 3: Wall time comparison for DecQN and REValueD. Each task was ran for 500k environment interactions (100k updates), and the mean ($\pm$ variance) is reported in time (minutes) over 3 seeds.

| Task | DecQN | REValueD |
|------|-------|----------|
| walker-walk | $30.47 \pm 0.00$ | $37.05 \pm 0.32$ |
| dog-walk | $55.46 \pm 0.31$ | $69.55 \pm 0.32$ |

To expedite data collection, we adopt an Ape-X-like framework (Horgan et al., 2018), with multiple distributed workers each interacting with individual instances of the environment. Their interactions populate a centralised replay buffer from which the learner samples transition data to learn from. It is crucial to note that all the algorithms we compare (REValueD, DecQN, BDQ) use the same implementation, ensuring a consistent number of environment interactions is used for each algorithm for fair comparison.

**Environment details:** In Table 5 we show the state and sub-action space size of the DM Control Suite tasks we use in the experiments in Section 5. In particular, we can see that the humanoid and dog environments have large state and action spaces, each having 21 and 38 sub-action spaces, respectively. With 3 bins per sub-action space, this would correspond to $3^{21}$ and $3^{38}$ atomic actions. It is particularly challenging to solve these tasks given the interdependencies between sub-actions.

Table 4: Hyperparameters used for the experiments presented in Section 5.

| Algorithm | Parameters | Value |
|-----------|-----------|-------|
|  | Optimizer | Adam |
|  | Learning rate | $1 \times 10^{-4}$ |
|  | Replay size | $5 \times 10^5$ |
|  | n-step returns | 3 |
|  | Discount, $\gamma$ | 0.99 |
|  | Batch size | 256 |
| General | Hidden size | 512 |
|  | Gradient clipping | 40 |
|  | Target network update parameter, $c$ | 0.005 |
|  | Imp. sampling exponent | 0.2 |
|  | Priority exponent | 0.6 |
|  | Minimum exploration, $\epsilon$ | 0.05 |
|  | $\epsilon$ decay rate | 0.99995 |
| REValueD | Regularisation loss coefficient $\beta$ | 0.5 |
|  | Ensemble size $K$ | 10 |

Table 5: Details of the DM Control Suite environments used in the experiments presented in Section 5. Note that $N$ corresponds to the number of sub-action spaces in the FMDP.

| Task | $|\mathcal{S}|$ | $N$ |
|------|------|-----|
| Cartpole Swingup Sparse | 5 | 1 |
| Reacher Hard | 6 | 2 |
| Ball in Cup Catch | 8 | 2 |
| Finger Spin | 9 | 2 |
| Fish Swim | 24 | 5 |
| Cheetah Run | 17 | 6 |
| Walker Walk/Run | 24 | 6 |
| Quadruped Walk/Run | 78 | 12 |
| Humanoid Stand/Walk/Run | 67 | 21 |
| Dog Walk/Trot/Run | 223 | 38 |

Table 6: Asymptotic results for REValueD with varying $\beta$ across various DM Control Suite tasks. We report the mean $\pm$ standard error over 10 seeds.

| Task | $\beta$ | | | | |
|---|---|---|---|---|---|
| | 0 | 0.25 | 0.5 | 0.75 | 1 |
| Finger-Spin | **947.80 $\pm$ 10.6** | 847.15 $\pm$ 20.3 | 905.86 $\pm$ 16.1 | 849.26 $\pm$ 20.3 | 927.39 $\pm$ 25.9 |
| Walker-Run | 755.72 $\pm$ 3.25 | 750.91 $\pm$ 2.95 | 761.74 $\pm$ 2.30 | **768.60 $\pm$ 1.62** | 758.87 $\pm$ 3.66 |
| Cheetah-Run | 756.71 $\pm$ 20.1 | 802.80 $\pm$ 18.3 | **828.60 $\pm$ 25.9** | 814.62 $\pm$ 19.5 | 792.52 $\pm$ 16.9 |
| Quadruped-Run | 885.91 $\pm$ 5.88 | 910.16 $\pm$ 8.86 | 924.52 $\pm$ 3.92 | 922.57 $\pm$ 12.3 | **927.21 $\pm$ 5.35** |
| Dog-Walk | 838.81 $\pm$ 14.3 | 870.75 $\pm$ 9.99 | 886.73 $\pm$ 7.38 | **895.54 $\pm$ 9.94** | 880.50 $\pm$ 9.96 |

## D  SENSITIVITY TO $\beta$

Whilst we did not tune $\beta$ for the results presented in Section 5, here we analyse how sensitive REValueD is to the value of $\beta$. Recall that in Equation (4.5) $\beta$ controls how much the regularisation loss contributes to the overall loss. In Table 6 we present asymptotic results for various DM Control Suite tasks using a varying $\beta$ value. Note that we performed the same number of updates per task as in Figure 1.

We can see that REValueD is reasonably robust to $\beta$, typically with the best performing $\beta$ value lying somewhere between 0.5 and 0.75. In the finger-spin task, we see that a $\beta$ value of 0 offers the best performance. Intuitively this would make sense, since there are only two sub-action spaces it is less likely that credit assignment would be an issue. Similarly in walker-run we see that there is not much difference in performance for the tested $\beta$ values. Much like finger-spin, this can be attributed to the fact that walker-run is a reasonably easy task and has only 6 sub-action spaces. Consequently, credit assignment may not have a major impact on the performance. In the dog-walk task we see that a $\beta$ value of 0 performs significantly worse, with the best performing $\beta$ values being 0.5 and 0.75. This further demonstrates the benefits of the regularisation loss, particularly in tasks with large $N$, whilst also showing the robustness to the value of $\beta$.

## E  FURTHER RESULTS FOR VARYING $n_i$

In this Section we present further results for varying $n_i$ in more DM control suite tasks. The results can be found in Figure 5. We observe that as $n_i$ increases, REValueD continues to exhibit better performance compared to DecQN and BDQ. However, it's worth noting that in the cheetah-run task, as $n_i$ surpasses 75, the performance difference becomes less pronounced.

## F  FURTHER RESULTS IN STOCHASTIC ENVIRONMENTS

In Figure 6 we present more comparisons for stochastic environments. We see that the results remain similar to those observed in Section 5, with the exception of the stochastic state variant of Quadruped-Run, where there is not a significant difference between REValueD and DecQN.

## G  TRAINING STABILITY

To investigate how this variance reduction influences the stability of the training process, we present in Figure 7 the Conditional Value at Risk (CVaR) for the detrended gradient norms, as suggested by Chan et al. (2019). The CVaR provides information about the risk in the tail of a distribution and is expressed as

$$\text{CVaR}(g) = \mathbb{E}\left[g|g \geq \text{VaR}_{95\%}(g)\right] \ .$$

Here, $g$ stands for the detrended gradient norm and VaR (Value at Risk) represents the value at the 95th percentile of all detrended gradient norm values. We perform detrending of the gradient norm by calculating the difference in the gradient norm between two consecutive time steps, that

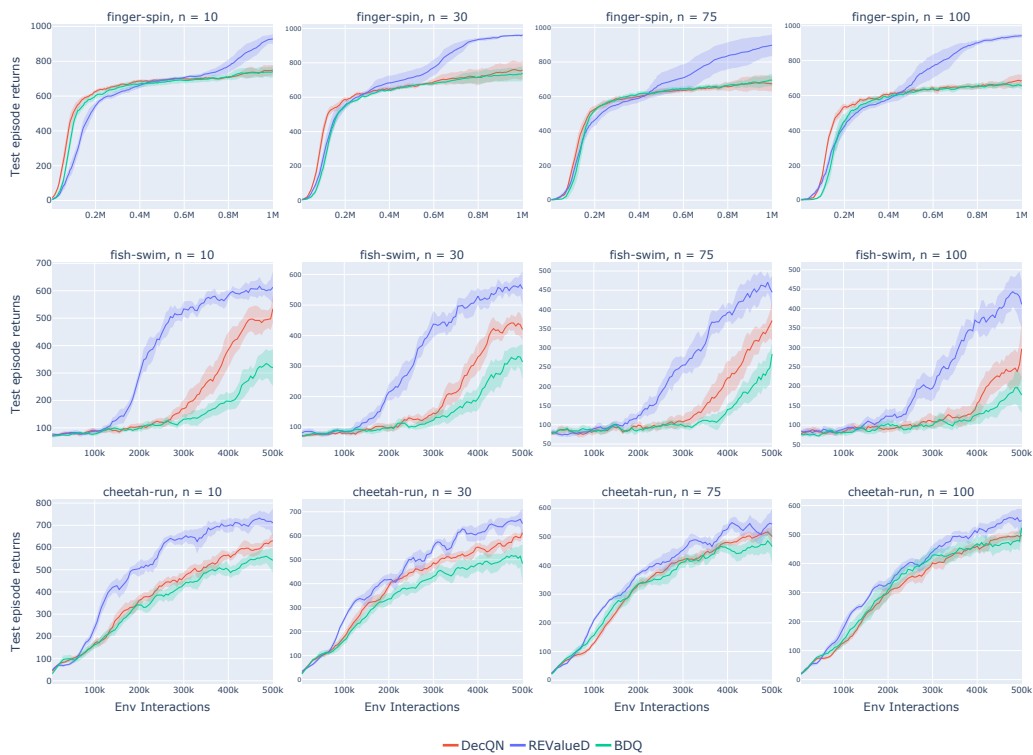

Figure 5: Further results assessing how the performance of DecQN, BDQ and REValueD are affected by increasing the size of each sub-actions space. $n$ corresponds to the size of the sub-action space, *i.e.* $|\mathcal{A}_i| = n$ for all $i$. The solid line corresponds to the mean of 10 seeds, with the shaded area corresponding to a 95% confidence interval.

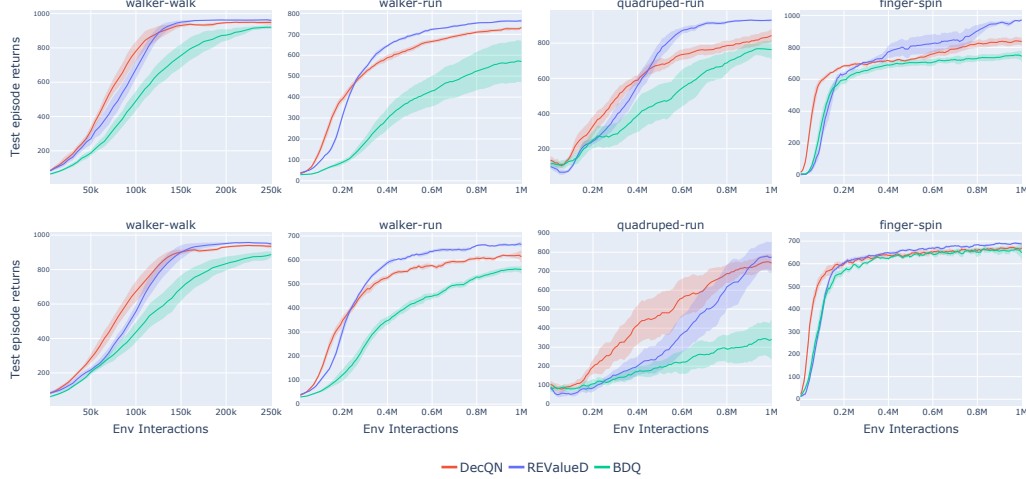

Figure 6: Stochastic environment tasks. In the top row we added Gaussian white noise ($\sigma = 0.1$) to the rewards, whilst in the bottom row we added Gaussian white noise to the state.

is, $g_{t+1} = |\nabla_{t+1}| - |\nabla_t|$, where $\nabla$ denotes the gradient. As shown in the Figure, employing the average of the ensemble for the learning target in REValueD considerably lowers the CVaR for the tasks under consideration.

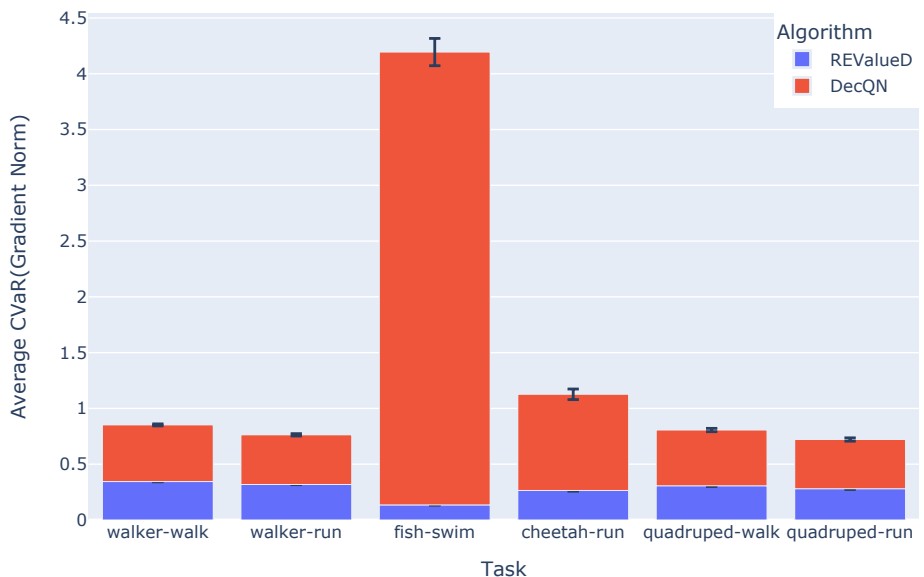

Figure 7: Conditional Value at Risk (CVaR) for the gradient norms (which have been detrended) for both DecQN and REValueD, evaluated on six tasks from the DeepMind Control Suite. The CVaR values presented here are the average across ten separate runs, and the figure includes both the mean values as well as error bars representing the standard error for each.

## H    CONTRIBUTION OF THE REGULARISATION LOSS IN A TABULAR FMDP

In this Section we design a simple FMDP encompassing $N$ sub-action spaces, each containing $n$ sub-actions. The FMDP includes one non-terminal state, with all actions resulting in a transition to the same terminal state. We have assumed that each sub-action space has a single optimal sub-action. A reward of $+1$ is obtained when all optimal sub-actions are selected simultaneously, whilst a reward of $-1$ is given otherwise.

For these experiments, we randomly initialise the utility values for every sub-action across all sub-action spaces using a Uniform$(-0.1, 0.1)$ distribution, with the exception of the optimal sub-action in the $N$th sub-action space, which is initialised with a value of $+1$. Actions are then chosen following an $\epsilon$-greedy policy ($\epsilon = 0.1$). After this, an update is performed on the transition data and we record the frequency at which the optimal sub-action in the final sub-action space has been taken. Since the utility values have been randomly initialised, the probability of jointly selecting the first $N-1$ optimal sub-actions is approximately $\frac{1}{n^{N-1}}$, which means that most of the transitions are likely to result in a reward of $-1$. Under the DecQN loss, performing an update on a transition with a reward of $-1$ will result in the value of the optimal sub-action in the $N$th sub-action space being decreased, despite it being initialised at an optimal value. We aim to show that using the regularisation loss from Equation (4.4) mitigates the effect that the transition has on the value of the optimal sub-action.

Figure 8 shows the mean frequency of optimal $N$th sub-action selection for both REValueD and DecQN. The mean is averaged over 1000 runs, and the shaded region represents a 95% confidence interval. A clear observation from this Figure is that, as both $N$ and $n$ increase, REValueD starts to outperform DecQN. This result underscores the efficacy of the additional regularisation loss in

reducing the impact of the sub-optimal sub-actions from the first $N - 1$ dimensions on the value of the optimal sub-action in the $N$th dimension.

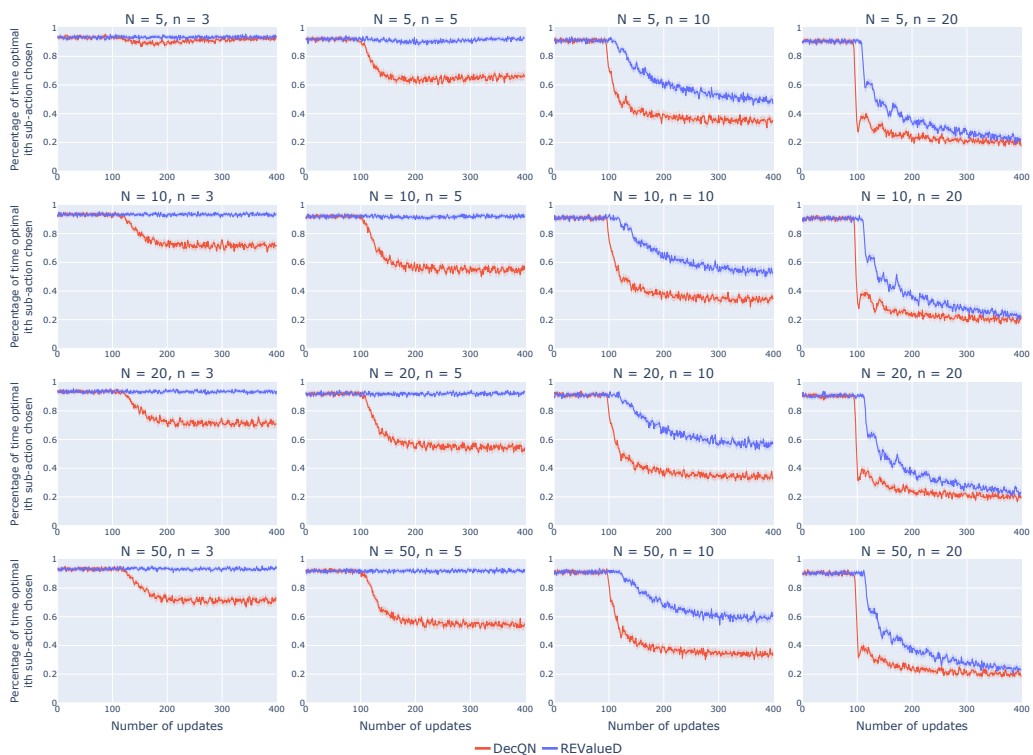

Figure 8: This figure presents the outcomes from the tabular FMDP experiment (see Section 5). The $x$-axis represents the number of model updates performed, whilst the $y$-axis signifies the frequency with which the optimal $N$th sub-action was selected, expressed as a percentage. The solid line corresponds to the mean performance over 1000 trials, and the shaded region corresponds to a 95% confidence interval.

## I FUNCTIONAL FORM OF THE REGULARISATION WEIGHTS

In Equation (4.4) we define the weighting as $w_i = 1 - \exp(-|\delta_i|)$. The motivation for this form was that, for large differences, we want the weight to tend to 1, and for small differences be close to 0. In this Section we look to assess how the performance of REValueD is affected when using a different function to define the weights that also satisfies this property. In Figure 9 we compare to a quadratic function of the form $w_i = \min(\delta_i^2, 1)$. The results can be found in Figure 10. We can see that generally speaking, the exponential weighting function offers a better performance in terms of sample efficiency, although both reach roughly the same asymptotic performance. The explanation for this can likely be attributed to the functional forms, demonstrated in Figure 9, where the quadratic function will return a weight of 1 for any $|\delta_i| \geq 1$. This leads to regularising the utility functions too harshly. It is possible that a temperature parameter $c$ could be incorporated to scale $\delta_i$, such that we now have $\delta_i' = \delta_i/c$ as the input to the weighting function, but an analysis of this is beyond the scope of this paper.

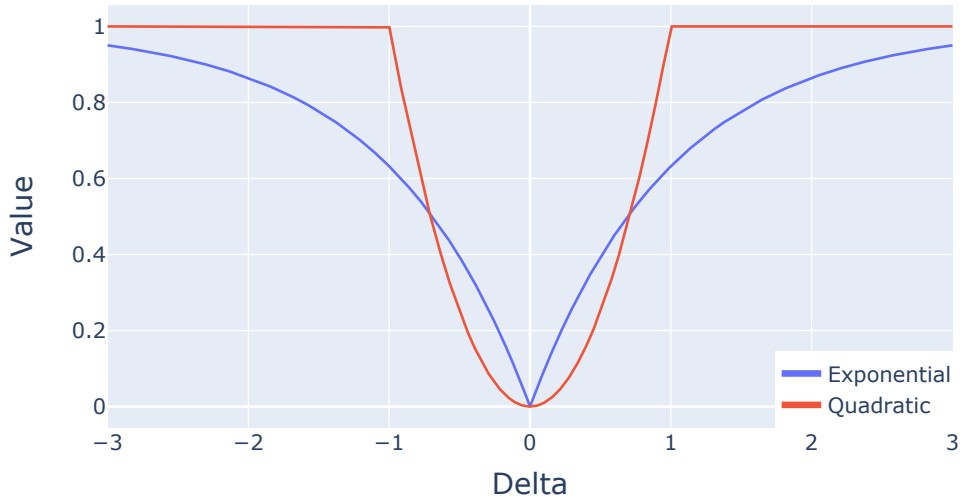

Figure 9: Here we show the plots of the exponential weight function and the quadratic weight function.

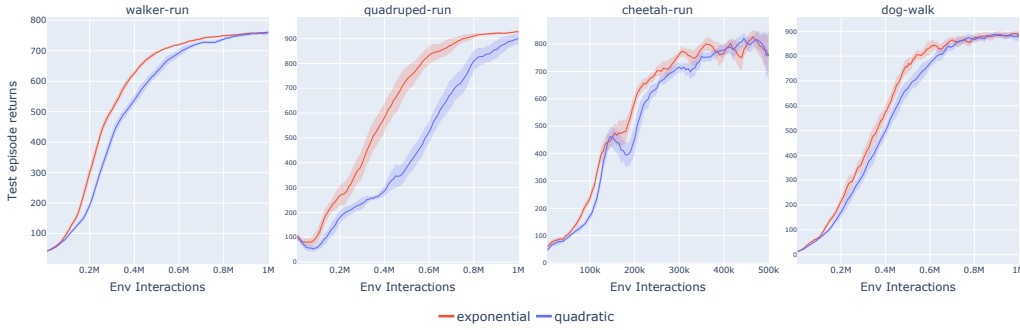

Figure 10: This Figure compares the performance of REValueD using the definition of the weights from Equation (4.4) (exponential) vs. a quadratic function. The solid line corresponds to the mean performance over 10 seeds, and the shaded region corresponds to a 95% confidence interval.

## J  METAWORLD

In this Section we offer further comparisons between DecQN and REValueD in some manipulation tasks from MetaWorld (Yu et al., 2020). The results can be found in Figure 11. We can see that in these tasks REValueD achieves stronger asymptotic performance than DecQN, whilst also generally having smaller variance. This demonstrates the efficacy in domains other than locomotive tasks.

## K  FURTHER COMPARISONS WITH ENSEMBLED BASELINES

For a more like-to-like comparison between REValueD and BDQ/DecQN-Sum, we have compared REValueD with variants of these baselines equipped with an ensemble – note that we have compared REValueD with DecQN+Ensemble in Table 1. The results for select environments can be found in

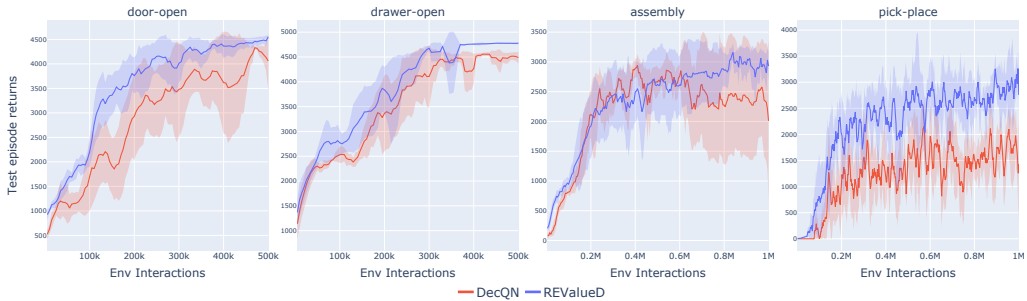

Figure 11: Here we compare the performance of DecQN and REValueD for discretised variants of manipulation tasks from MetaWorld. The solid line corresponds to the mean of 10 seeds, with the shaded area corresponding to a 95% confidence interval.

Figure 12. We can see that the results for BDQ remain largely unchanged, with the exception of finger-spin with a bin size of 10, where BDQ+Ensemble now attains a similar performance to that of REValueD. For DecQN-Sum+Ensemble we note that the performance in finger-spin is improved by equipping an ensemble, with the performance achieving similar levels to REValueD for $n = 30, 75, 100$. However, for the other tasks the performance is still largely unchanged.

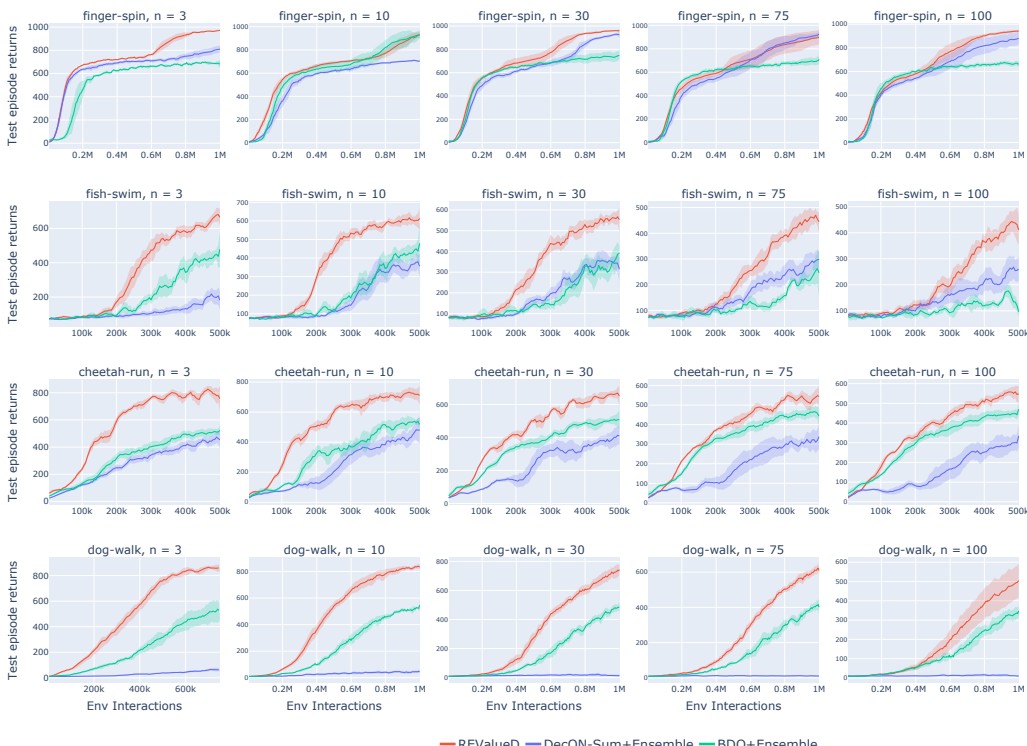

Figure 12: Here we compare the performance of REValueD with BDQ+Ensemble and Sum-DecQN+Ensemble on discretised variants of the DM control suite tasks with varying bin sizes. The solid line corresponds to the mean of 10 seeds, with the shaded area corresponding to a 95% confidence interval.

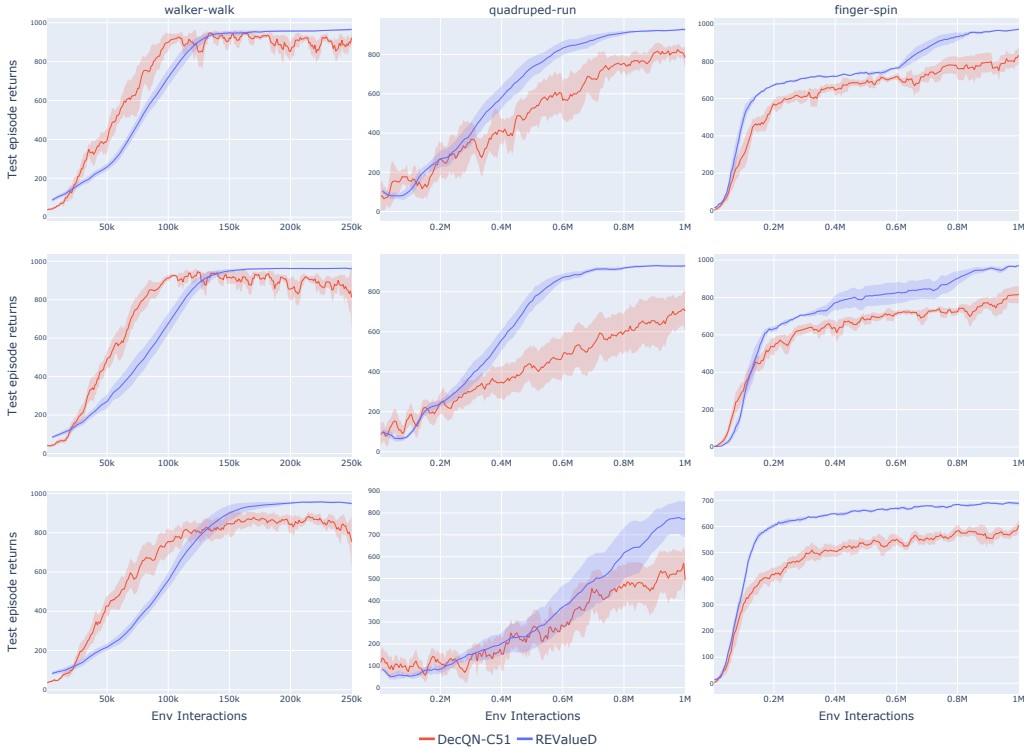

Figure 13: Comparison of DecQN-C51 and REValueD. The top row are non-stochastic DM control suite tasks; whilst the middle and bottom rows correspond to the same environments with Gaussian white noise ($\sigma = 0.1$) added to the rewards and states, respectively. The solid line corresponds to the mean of 10 seeds, with the shaded area corresponding to a 95% confidence interval.

## L  DISTRIBUTIONAL CRITICS

As noted in Section 6, a distributional perspective to learning can help deal with uncertainty induced by exploratory sub-actions. To that end, similar to the work in Appendix I of Seyde et al. (2022), we compare our method, REValueD, to a variant of DecQN which uses a distributional critic based on C51, introduced by Bellemare et al. (2017). Rather than learning an expected utility value for each sub-action, the critic now learns a distribution over values for each sub-action. The decomposition now proceeds at the probability level by averaging over logit values $\ell = \sum_{i=1}^{N} \ell_i / N$.

We observe the results of the comparison between DecQN-C51 and REValueD in Figure 13, with comparisons being conducted in selected DM Control Suite environments and their stochastic variants. We observe that, even when equipped with a distributional critic, REValueD maintains a superior performance in all of the tasks.

