# OpenReview forum: "REValueD: Regularised Ensemble Value-Decomposition for Factorisable Markov Decision Processes"
_ICLR.cc/2024/Conference — ICLR 2024 poster_

### Official Review · Reviewer_Athy · 2023-10-22

**Soundness:** 3 good
**Presentation:** 3 good
**Contribution:** 3 good
**Rating:** 6
**Confidence:** 4

**Summary:**

This paper studies a factorized Q-learning approach to solve continuous control problems. The approach builds on the recent Decoupled Q-Networks agent and makes two extensions: (1) critic ensembling to mitigate value variance, (2) a regularization objective to mitigate credit assignment issues stemming from exploratory sub-actions. The resulting REValueD agent improves learning efficiency as measured by gradient step count on common DeepMind Control Suite tasks.

**Strengths:**

-	The ensembling and regularization objective are natural extensions to the DecQN agent
-	The resulting REValueD agent yields strong performance on the tasks being studied
-	Ablations on individual components highlight the benefit of each addition in isolation
-	The tabular FMDP experiment in Appendix G is a nice addition
-	The paper is well-motivated, structured and written

**Weaknesses:**

-	The results focus on DeepMind Control Suite tasks from **proprioceptive inputs**. The primary baseline additionally evaluated on MetaWorld, Control Suite from vision, and an Isaac Gym-based locomotion task with a single set of hyperparameters (some minor variations). The results presented here would benefit from additional evaluations on more diverse environments.
-	**Control Suite performance** (Figure 1): the results would benefit from displaying reference performance of a strong recent continuous actor-critic agent. The benchmark results (throughout) are provided with “Gradient steps” on the x-axis, which deviates from the conventional “Time steps” metric. Readability would be improved by adding Figure 1 with “Time steps” to the Appendix and providing the necessary conversion values between “Gradient steps" and "Time steps” (e.g. gradient steps per time steps ratio).
-	**Increasing discretization** (Figure 2 & 4): the robustness of REValueD over increasing discretizations is impressive. Figure 6 of the DecQN paper showed approximately the same performance on both Dog-Walk and Finger-Spin when using 3 vs 21 bins – do you have an intuition for where this difference is coming from (Figure 6 of DecQN yields ~950 on Finger-Spin and ~850 on Dog-Walk for 21 bins). Is this in part due to the hyperparameter modifications over the original DecQN agent mentioned in Appendix B?
-	**Stochastic environments** (Figure 3 & 5): the DecQN paper also introduced a distributional DecQN agent based on the C51 critic in Appendix I with experiments in stochastic environments in Appendix J. Distributional critics can be viewed as an alternative method to ensembling in accounting for variability resulting from exploratory sub-actions, potentially side-stepping the computational overhead of explicit ensembles. Discussion and/or experimental comparison between DecQN+C51 or a distributional version of REValueD and ensembled REValueD would further strengthen the paper (this was briefly mentioned in the conclusion).
-	A potential downside of ensembles is the **computational overhead** required for training. It would be interesting to see how the plots compare when plotting time on the x-axis. While Appendix B notes that there is “minimal slowdown” for relatively small ensemble sizes, it would be helpful to quantify this.

**Questions:**

-	How would REValueD ensembling compare to distributional DecQN?
-	How was the ensembling set up to ensure efficiency / what is the computational overhead?
-	What was the benefit of altering certain hyperparameters (exploration decay, Polyak averaging) and are these sufficient to explain the performance mismatch between DecQN in Figure 2 & 4 (REValueD) vs Figure 6 (DecQN)?

---

> ### Author Response · Authors · 2023-11-13
> **Response to reviewer Athy**
>
> **Further experimental results**
>
> Thank you for your valuable feedback on the scope of our evaluations. We agree that expanding the range of environments tested would provide a more comprehensive understanding of REValueD's effectiveness. The current focus on DeepMind Control Suite tasks from proprioceptive inputs was a starting point to establish the method's performance. We acknowledge the importance of testing in more diverse environments as these could offer insights into the versatility and robustness of our approach under varying conditions. As such, we have added results for DecQN and REValueD in a selection of Metaworld tasks (Appendix J). We see that REValueD still maintains superior performance over DecQN in these tasks, generally with smaller variance.
>
> **Control suite performance**
>
> For the metric used in our results, we agree that using 'Time steps' instead of 'Gradient steps' on the x-axis would align better with conventional standards and enhance readability. We have made the necessary adjustments to the figures to display 'Time steps' (the ratio was 5 env steps per update for all non-quadruped environments, in which case we used 10).
>
> Regarding the inclusion of a continuous actor-critic agent as a baseline, our focus is on discrete action spaces in this study. Particularly in the context of DecQN and REValueD, the focus was driven by the specific challenges and characteristics of high-dimensional discrete action spaces. Since continuous control isn't the primary focus of our work, we believe that adding a continuous actor-critic baseline might not directly contribute to the comparative analysis intended for our research. However, we appreciate the suggestion and will consider it for potential future studies where a comparison with continuous control methods would be more relevant.
>
> **Increasing discretisation and sensitivity to hyper-parameters**
>
> Thank you for pointing out this discrepancy between the performance in Fig 6 of Seyde et al. and Figs 2,4  of our paper.  In our revised paper, we have now plotted figures with environmental interactions on the x-axis which allows a more direct comparison.  For dog-walk, after 1M environment interactions the score in Seyde et al. is ~650 and in ours it is ~550.  This discrepancy appears to be the result of poor performance on a single seed in which our DecQN was unable to learn past a score of ~50.  This can be seen in Fig 2 in which the variance of scores is uncharacteristically high.  Removing this seed from our results, we average ~650, more in line with Seyde et al.
>
> For finger-spin, after 1M environment interactions the score in Seyde et al. is ~850 and in ours it is ~750.  In this instance, we don’t believe this is the result of an “outlier” seed (variance in scores is low), rather this may be due to slight difference in implementation and/or hyperparameter adjustments as you have suggested.
>
> As for the reason for the differences in changing the hyper-parameters, the simple answer is that when we initially started experimenting with REValueD, this is how our existing code base was implemented. We did test to see whether they effected performance, but there was no significant changes between the implementations.
>
> **Computational overhead**
>
> Your suggestion to include timing results is indeed pertinent and would offer a more comprehensive understanding of the practical implications of using ensembles, particularly in terms of training time.
>
> In our manuscript, though we cited a paper which claimed 'minimal slowdown' for relatively small ensemble sizes, we agree that providing empirical data to quantify this would be significantly beneficial. To address this, on the same machine we ran 3 seeds each for walker-walk and dog-walk, for DecQN and REValueD. Each seed ran for 500k env interactions (100k updates). We have included the results in Table 3, Appendix C. We observe around a 25\% slow down using REValueD, but it is important to consider that REValueD is more sample efficient than DecQN and so less environment interactions are needed to achieve a superior performance. We have also changed the wording from 'minimal' to 'minor'
>
> **Comparison with distributional DecQN**
>
> You make an important point about distributional critics being an alternative to ensembles in addressing variability due to exploratory sub-actions, and their potential advantage in computational efficiency.
>
> Whilst we briefly mentioned the potential of distributional reinforcement learning in our conclusion, we agree that incorporating a more detailed discussion and experimental comparison would strengthen our paper. We have added a comparison between DecQN with a distributional critic in Appendix L, finding that REValueD maintains its superior performance.

---

> > ### Comment · Reviewer_Athy · 2023-11-20
> > **Response to rebuttal**
> >
> > Thank you for your replies! The added experiments and evaluations make the paper more well-rounded. I still think that a conventional continuous control actor-critic method would provide a valuable reference for readers to better assess the strong performance of the method. All in all a nice approach with strong results on the tasks considered!

---

> > > ### Author Response · Authors · 2023-11-21
> > > **thanks**
> > >
> > > Thanks for recognizing the improvements in our paper! We've noted your suggestion for future work. Given the enhancements, could you consider revising the review score? Your further input would be valuable.

---

### Official Review · Reviewer_bJ1N · 2023-10-25

**Soundness:** 3 good
**Presentation:** 3 good
**Contribution:** 3 good
**Rating:** 6
**Confidence:** 3

**Summary:**

The paper proposes REValueD to enable Q-learning algorithms on tasks with high-dimensional discrete action space. With the same concept of value decomposition, REValueD improves DecQN by mitigating the target variance with an ensemble of critics and mitigating the effects of exploratory actions with a regularization loss. Experiments on the DeepMind Control Suite tasks show that REValueD consistently outperforms DecQN and another baseline.

**Strengths:**

1.	This paper is well-written and well-structured. Although the ideas of the ensemble and regularization is incremental, they are straightforward and make sense.
2.	The theoretical analysis shows that REValueD could reduce the target variance of DecQN while maintaining the expectation of the target difference of DecQN unchanged with the help of the ensemble technique.
3.	The experiments demonstrate the effectiveness of the proposed method and ablation studies are given to further validate each component of REValueD.

**Weaknesses:**

1.	In the experiments, only two baselines (DecQN and BDQ) are compared although the authors list several works in the related works.
2.	An introduction and analysis of the action pace and sub-action space in humanoid and dog tasks may help the readers understand the setting and motivation more clearly.

**Questions:**

1.	In Equation (3.1), the value-decomposition form is the direct sum operator and REValueD follows this form, why do the authors not use other value-decomposition forms such as weighted sum?
2.	Could the authors explain more about the design of |δi|? Why “for large |δi|, the reward/next state is likely influenced by the effect of other sub-actions.”?

---

> ### Author Response · Authors · 2023-11-13
> **Response to reviewer bJ1N**
>
> **Choice of value decomposition form**
>
> In our study, we focused on the DecQN value-decomposition, which utilises the mean operator. This choice was driven by our specific interest in exploring and building upon the DecQN framework. While other forms of value-decomposition, such as the  weighted sum, offer interesting alternatives, our study aimed to investigate the implications of the DecQN approach in particular. We acknowledge that other decomposition forms could lead to different  insights and outcomes, and agree that exploring these alternatives would be a valuable avenue for future work.
>
> **Description of action and sub-action spaces for dog/humanoid**
>
> We have added a brief description of the action and sub-action spaces in the humanoid/dog in Appendix C, under environment details. We have elaborated on the unique challenges posed by the humanoid and dog tasks in the DeepMind Control Suite, such as the large number of joints (sub-actions) that need to be controlled and the intricate interdependencies among these joints. This will provide readers with a clearer understanding of the setting and underline the motivation behind our approach, particularly highlighting why these tasks are well-suited for evaluating the performance of REValueD and the implications of our method on managing such complex action spaces.
>
>
> **Design of $|\delta_i|$**
>
> Regarding the design of $|\delta_i|$ and its implications, we appreciate the opportunity to clarify this aspect. The concept of $|\delta_i|$ is central to our approach in handling the credit assignment issue. The rationale behind the design is that a large $|\delta_i|$ represents the difference between the current and target utility values for a sub-action. Therefore, a large $|\delta_i|$ might indicate significant impact from other sub-actions or external factors. This is because in complex environments, especially with multiple sub-actions, the effect of an individual sub-action is often intertwined with others. Therefore, when $|\delta_i|$ is large, it indicates that the reward or the state transition is likely not solely determined by the sub-action in question, but rather is influenced by a combination of several sub-actions.
>
> **Baselines**
>
> Whilst we only compared to two main baselines in the main paper, we have added in the revised version comparisons to BDQ equipped with an ensemble, as well as DecQN equipped with a distributional critic (DecQN-C51), in Appendix K and L, respectively.

---

> > ### Author Response · Authors · 2023-11-21
> >
> > Reviewer bJ1N - we've carefully considered and responded to your comments on our submission, particularly focusing on the choice of value-decomposition form, the design of |δi|, and expanding our baselines. Your insights were invaluable in guiding these revisions. Could you please take a moment to review our updated response and let us know if it addresses your concerns? We're eager to hear your thoughts on these changes.

---

> > > ### Comment · Reviewer_bJ1N · 2023-11-22
> > >
> > > The reviewer appreciates the authors' response. My main concerns are addressed including the baselines and descriptions of the action space. I would like to maintain my score.

---

### Official Review · Reviewer_fnTY · 2023-10-28

**Soundness:** 1 poor
**Presentation:** 3 good
**Contribution:** 2 fair
**Rating:** 5
**Confidence:** 4

**Summary:**

This paper studies RL in large discrete action space with factorization structure. Building upon the value decomposition idea proposed in past work, this paper presents some theoretical analysis which motivates the use of an ensemble of critics to control the variance as well as a new regularization term for better coordination over sub-actions. Experiments on discretized version of DeepMind control suite shows competitive performance.

**Strengths:**

- Writing is clear.
- Strong empirical results showing better performance than the baselines considered.
- Ablation studies show the effect of regularization term and ensemble size, which is helpful for future readers who may want to adopt the proposed method.

**Weaknesses:**

- The theoretical analysis on "DecQN target estimation bias and variance" should be made more rigorous, and some of the conclusions in this paper seems to contradicts previous results in an uncited related work.
- The justification for the "Regularized value-decomposition" seems more of a heuristic rather than a formal justification.
- The authors should consider adding a few missing baselines to the main experiment in Figure 1 for a fairer comparison

These are elaborated in Questions below.

**Questions:**

- Bias and Variance
  - In Tang et al. 2022, the authors showed that using Q decomposition may **increase bias** and **decrease variance**. In theorem 1 of this paper, the authors claim using Q decomposition may **decrease bias** and **increase variance**. These results seem to contradict each other. Could you clarify any difference in the analyzed settings?
  - Eqn (3.1) is valid to write since we are defining the parameterization of the Q-function $Q_{\theta}$, however, in Eqn (4.2) for the term related to $U_i^{\pi_i}$, the decomposition of the true Q-function $Q^{\pi}$ might not exist, as discussed in Tang et al. 2022. Could you elaborate what assumptions is the current analysis operating under?

- Fairer comparisons with baselines
  - The proposed method appears to be modifying DecQN to incorporate (1) ensemble and (2) regularization. In Fig 1, REValueD should be compared to ensembled DecQN and ensembled BDQ for a more "apples-to-apples" comparison.

- Other questions:
  - In Fig 2, why does BDQ perform well on this task but not the tasks in Fig 1? Would BDQ with ensemble perform even better?
  - Minor presentation suggestion: in Table 2, can you bold best results for each task to make it easier to read?

- References:
  - Tang et al. Leveraging Factored Action Spaces for Efficient Offline Reinforcement Learning in Healthcare. NeurIPS 2022.
https://openreview.net/forum?id=wl_o_hilncS

---

> ### Author Response · Authors · 2023-11-13
> **Response to reviewer fnTY**
>
> **Comparison with Tang et al. 2022**
>
> Thank you for bringing this piece of research to our attention, which we have now included in our Related Work section.  While both studies operate in the realm of Q-function decomposition within factored action spaces, there are two key distinctions with respective to their theoretical analyses.
>
> First and foremost, our analysis focuses on the bias/variance of the error between the true and (function) approximated Q-values, whereas Tang et al. focus on the fundamental properties of the Q-values themselves. The outcomes of our respective analyses reflect how these related, but distinct, attributes change when moving from the standard Q-learning approach to one based on value-decomposition, and hence these analyses can be thought of as complementary rather than contradictory.
>
> Second, there is a minor difference in the value-decomposition itself.  We use the decomposition proposed by Seyde et al., which involves taking a _mean_ of utility values to approximate the global Q-values, whereas Tang et al. take the _sum_ of utility values.  For completeness, we have added a section in Appendix B that analyses this sum variant under our framework, and we find this time that both the bias and variance are higher.  We could of course mirror Tang et al’s. analysis using the mean instead of the sum, but this is beyond the intended scope of our paper which is interested in the error as stated above.
>
> In summary, the apparent discrepancies concerning variance can be attributed to varying methodologies and distinct theoretical frameworks, which when taken together provide complementary perspectives and insights into Q-learning in FMDPs using value-decomposition.
>
> **Q-value decomposition**
>
> Our analysis assumes that the Q-function can be decomposed for practical approximation, an approach extended to the true Q-function in Equation (4.2). While, as Tang et al. 2022 highlight, such decomposition may not perfectly align with the true Q-function in practice, we posit that it offers a close enough approximation for the purposes of reinforcement learning, where exact representations are often unattainable. This assumption underpins our theoretical framework and the development of REValueD. The strong empirical results, acknowledged by the reviewer, provide further evidence that this decomposition approach is adequate in many practical tasks.
>
> **Performance of BDQ**
>
> Thank you for pointing out that a more like-for-like comparison of BDQ would be to compare to BDQ with an ensemble. In Appendix K we have added some comparisons with BDQ-ensemeble in the environments where the bin size is 3, and in the variants with larger bin sizes, and find that the performance remains largely unchanged when equipping BDQ with an ensemble.
>
> As for the performance of BDQ in Figures 1 vs. 2, it is important to note that the x-axis scale (also note as per request of reviewer Athy we have changed x-axis scale from updates to environment interactions) changes slightly between Figures 1 and 2 for the dog-walk task. In Figure 1 we cut the figure off after 750k environment interactions, at which point BDQ obtains a score of ~500, whereas BDQ in Figure 2 is achieving approximately this score after 1M env interactions for $n = 10, 30$, so it is taking more environment interactions to achieve the same score when using a higher bin size, though the performance does not diminish as quickly as DecQN when using a higher bin size.
>
> **Presentation suggestions**
>
> Regarding your suggestion for Table 2, we agree that bolding the best results for each task would significantly enhance readability and quick comprehension of the table. We have incorporated this in the revised version of the manuscript.

---

> > ### Author Response · Authors · 2023-11-21
> > **Does our reply**
> >
> > Reviewer fnTY- we've responded to your comments with a focus on clarifying how our work complements Tang et al. 2022, addressing a crucial point you had raised. Could you please review our revisions and let us know if they resolve your concerns? Your additional feedback would be highly valued.

---

### Author Response · Authors · 2023-11-13
**Additional experiments and revisions**

We would like to thank all reviewers again for their feedback and comments.

We have tried to address all comments individually, and we have uploaded a revised version of the paper with new text/captions in blue.

If there are any remaining questions with regards to the original paper or the revised version, please do get in touch. Thanks again!

---

### Author Response · Authors · 2023-11-20
**Please kindly provide feedback**

As the discussion period comes to a close, we kindly request your feedback regarding our rebuttal. We value your input on whether our response effectively addressed your concerns and if there are any remaining questions or points you would like us to elaborate on for further clarity.

---

### Meta-Review · Area_Chair_Maxj · 2023-12-20

**Metareview:**

The paper presents an approach to handling large discrete action spaces in reinforcement learning. The reviewers appreciated the method's motivation, simplicity and improved performance compared to baselines and ablation studies. While there were concerns about the scope of the evaluation in terms of baselines and domains, the reviewers agreed the paper's promising results outweigh these issues. For these reasons, I recommend acceptance.

**Justification For Why Not Higher Score:**

Limited scope of contribution.

**Justification For Why Not Lower Score:**

Motivation, simplicity and improved performance.

---

### Decision · Program_Chairs · 2024-01-16

Accept (poster)